# Pair2Scene: Learning Local Object Relations for Procedural Scene Generation

**Xingjian Ran** [1]   **Shujie Zhang** [2]   **Weipeng Zhong** [3]   **Li Luo** [1 4]   **Bo Dai** [1]

Project Page

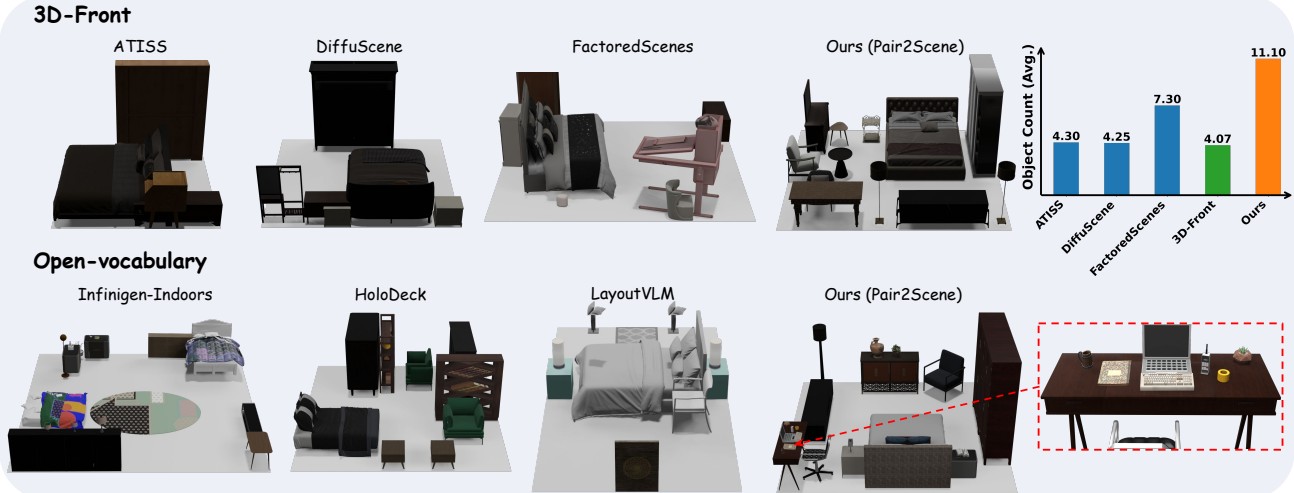

*Figure 1.* **Scene Generation Results.** (Top) When trained solely on curated 3D-Front data (avg. 4.07 objects for bedroom), learning-based methods remain confined to its limited distribution. In contrast, Pair2Scene beyond dataset distribution by leveraging local object relations to procedurally generate more complex scenes. (Bottom) When trained on full proposed 3D-Pairs dataset, Pair2Scene maintains superior spatial logic over procedural or LLM-based baselines, producing richer, more realistic environments.

## Abstract

Generating high-fidelity 3D indoor scenes remains a significant challenge due to data scarcity and the complexity of modeling intricate spatial relations. Current methods often struggle to scale beyond training distribution to dense scenes or rely on LLMs/VLMs that lack the ability for precise spatial reasoning. Building on top of the observation that object placement relies mainly on local dependencies instead of information-redundant global distributions, in this paper, we propose **Pair2Scene**, a novel procedural generation framework that integrates learned local rules with scene hierarchies and physics-based algorithms. These rules mainly capture two types of inter-object relations, namely *support relations* that follow physical hierarchies, and *functional relations* that reflect semantic links. We model these rules through a network, which estimates spatial position distributions of dependent objects conditioned on position and geometry of the anchor ones. Accordingly, we curate a dataset **3D-Pairs** from existing scene data to train the model. During inference, our framework can generate scenes by recursively applying our model within a hierarchical structure, leveraging collision-aware rejection sampling to align local rules into coherent global layouts. Extensive experiments demonstrate that our framework outperforms existing methods in generating complex environments that go beyond training data while maintaining physical and semantic plausibility.

## 1. Introduction

The generation of high-fidelity, realistic 3D scenes is fundamental to the advancement of the game industry, embodied AI, virtual reality, and film production. Indoor 3D scenes are essentially composed of individual object entities and their spatial layouts, and recent advancements (Zhang et al., 2024; Xiang et al., 2024; Zhao et al., 2025; Lai et al., 2025) in generative modeling have already driven a rapid surge

---

[1]The University of Hong Kong [2]Tsinghua University [3]Shanghai Jiao Tong University [4]Shenzhen Loop Area Institute. Correspondence to: Bo Dai <bdai@hku.hk>.

*Proceedings of the 43rd International Conference on Machine Learning*, Seoul, South Korea. PMLR 306, 2026. Copyright 2026 by the author(s).

in the quality of object-level generation. However, realistic layout generation remains a significant bottleneck, where one of the primary challenges lies in the effective modeling and generating complex scenes. This challenge is rooted in a systemic data deficiency. Existing synthetic datasets (Fu et al., 2021; Hao et al., 2025) are often limited in complexity, focusing on specific domains, like large furniture or tabletops. Meanwhile, real-to-sim datasets (Zhong et al., 2025) suffer from noisy data distribution. Consequently, neither source can fully support the end-to-end training of models for generating complex and diverse scenes. Given these difficulties, the problem of how to efficiently represent the intricate relations in global scenes is critical, yet largely underexplored.

Beyond the data hurdles, existing approaches also face structural limitations in modeling complex scene distributions. Existing learning-based methods (Paschalidou et al., 2021; Yang et al., 2024a; Tang et al., 2024; Lin & Mu, 2024; Zhai et al., 2024) mainly rely on end-to-end training within a single dataset. Although these methods effectively capture the dataset's distribution, there are two primary limitations. First, the generative capacity of the model is inherently capped by the complexity of the training data, hindering its ability to synthesize novel or more intricate environments. Second, as the number of objects increases, the complexity of modeling the joint distribution rises disproportionately. Since every object is often treated as globally dependent on all others, the interaction between objects becomes increasingly difficult to learn. An alternative research thrust attempts to leverage the vast commonsense knowledge of language models to reason about scene layouts. While LLM/VLM-guided approaches (Zhou et al., 2024; Fu et al., 2024; Yang et al., 2024b; Aguina-Kang et al., 2024; Li et al., 2025; Deng et al., 2025) can produce diverse and semantically rich scenes, they are constrained by the limited spatial reasoning capabilities, often resulting in layouts that lack physical or semantic plausibility.

In this paper, we raise a question: *Is it truly necessary to model the global distribution of an entire scene?* We argue that the complexity of scene modeling can be significantly mitigated by *rethinking the scope of object interactions*. To this end, we propose **Pair2Scene**, a framework grounded in the observation that the placement of any given entity in a dense indoor environment is typically governed by only a small subset of proximal neighbors rather than the entire global context. By modeling local rules, which are more abundant and consistent across datasets, this decomposition effectively addresses data scarcity and facilitates seamless joint training. To formalize local rules, we define two primary categories of inter-object relations: support relations, which characterize the physical support dictated by gravity, and functional relations, which describe spatial distributions influenced by the semantic and functional links

between two objects. Based on these definitions, we build a data curation pipeline that aggregates local rules from multiple sources to construct our **3D-Pairs** dataset. Using object point clouds as input, our model learns the spatial distribution of object relations, thereby bridging the gap between fine-grained asset geometry and spatial distribution. Furthermore, to achieve global consistency and physical plausibility, Pair2Scene combines scene hierarchies, local layout learning, and physics-based algorithms to execute local distributions in a structured sequence while maintaining awareness of the overall scene configuration. This design effectively simulates a global spatial distribution, allowing Pair2Scene to surpass training data constraints and generate scenes of significantly higher complexity than the original data.

Our contributions are summarized as follows:

- We introduce Pair2Scene that leverages point cloud to learn and predict local spatial distributions, incorporating asset geometry into the placement. By strategically combining local rule learning with scene hierarchies and physics-based algorithms, our framework achieves scalability and alleviates data scarcity by extracting abundant local rules from diverse sources.
- We formally define a set of object relations as the foundation for local rules, encompassing support hierarchies and functional links. To support this representation, we develop a robust data curation pipeline, culminating in the 3D-Pairs dataset, which contains approximately 140,000 relation tuples.
- Extensive experiments show that Pair2Scene generates complex scenes with higher quality than existing methods in both physical plausibility and semantic alignment, and demonstrates the ability to generalize to out-of-distribution scenes, producing layouts of higher complexity than those in training data.

## 2. Related Work

### 2.1. Learning-based Scene Generation

Learning-based methods (Feng et al., 2023; Sun et al., 2025b; Ran et al., 2025; Feng et al., 2025) mainly focus on modeling scene distributions directly from scene datasets, typically treating scene synthesis as a sequence generation task. Among these, ATISS (Paschalidou et al., 2021) utilizes a Transformer-based model to achieve scene completion and generation by predicting object attributes. DiffuScene (Tang et al., 2024) introduces denoising diffusion models to the spatial domain, enhancing generation quality through iterative refinement in the object attribute space. To improve interactivity, InstructScene (Lin & Mu, 2024) enables local editing and arrangement via natural language instructions. Recent works like FactoredScene (Hsu et al., 2025) employ

program-based representations to learn procedural libraries for scene construction. Furthermore, LayoutVLM (Sun et al., 2025a) leverages multimodal alignment to enhance spatial reasoning through vision-language semantics. Despite their success in capturing the distribution of datasets, the generative capacity of these models is strictly bounded by the diversity of the training samples, and they struggle to scale as the object count rises, since the inter-object dependencies become increasingly intractable to model.

## 2.2. LLM/VLM-based Scene Generation

Other works (Fu et al., 2024; Aguina-Kang et al., 2024; Deng et al., 2025) utilize zero-shot reasoning and broad commonsense knowledge of LLMs/VLMs to bypass the requirement for extensive scene-specific training. GALA3D (Zhou et al., 2024) utilizes LLMs to generate initial layouts coupled with object-scene compositional optimization for multi-object generation. I-Design (Çelen et al., 2024) focuses on interactive design, integrating user preferences with generative models for real-time, incremental scene construction. HoloDeck (Yang et al., 2024b) generates detailed layout schemes via LLMs to create high-quality environments suitable for robotics simulation. Additionally, HSM (Pun et al., 2026) introduces a hierarchical scene model that organizes generation from room architecture to furniture groups to improve structural logic. While these approaches offer significant flexibility and semantic diversity, they predominantly rely on holistic scene modeling. This global perspective creates a reasoning bottleneck as scene complexity increases. The inherent spatial limitations of language models make it increasingly difficult to provide precise guidance for high-density environments.

## 3. Method

### 3.1. Problem Formulation

We define a 3D scene $\mathcal{S} = \{\mathcal{O}_1, \mathcal{O}_2, \ldots, \mathcal{O}_N\}$ as a structured collection of objects. Each object $\mathcal{O}_i$ is characterized by its geometric asset $\mathcal{A}_i$ and its oriented bounding box $\mathbf{B}_i = \{\mathbf{c}_i, \mathbf{s}_i, \mathbf{r}_i\}$, where $\mathbf{c}_i \in \mathbb{R}^3$ denotes the center coordinates, $\mathbf{s}_i \in \mathbb{R}^3$ the dimensions, and $\mathbf{r}_i \in \mathbb{R}^6$ the continuous rotation representation (Zhou et al., 2019).

#### 3.1.1. OBJECT RELATIONS AND SCENE SEQUENCE

To manage the complexity of dense environments, we decompose the scene into a sequence of local object interactions. To formalize these local rules, we define two primary relational archetypes that govern the spatial and semantic configuration of the scene:

**Support Relation** ($R_s$): Grounded in physical plausibility, this relation identifies an anchor object as the supporting surface (e.g., a table) that provides gravitational stability to

a dependent object (e.g., a laptop).

**Functional Relation** ($R_f$): This characterizes the semantic proximity between entities sharing a common surface, where the dependent object's placement is conditioned by its functional utility relative to an anchor (e.g., a keyboard placed relative to a laptop).

Based on these relations, a global scene is represented as an ordered sequence of relational tuples $\mathcal{S} = \{\mathcal{T}_1, \mathcal{T}_2, \ldots, \mathcal{T}_N\}$. Each tuple $\mathcal{T}_i$ is formulated as:

$$\mathcal{T}_i = \langle \mathcal{O}_{dep,i}, \mathcal{O}_{sup,i}, \{\mathcal{O}_{fnc,i}\}_{opt} \rangle, \quad (1)$$

where $\mathcal{O}_{dep,i}$ is the object to be generated, $\mathcal{O}_{sup,i}$ is the mandatory support anchor, and $\mathcal{O}_{fnc,i}$ is an optional functional anchor.

To ensure a valid generative dependency, the sequence must maintain causality: for any $i$-th tuple, the anchor objects ($\mathcal{O}_{sup,i}$ and $\mathcal{O}_{fnc,i}$) must either be the floor or have been previously instantiated as a dependent object $\mathcal{O}_{dep,j}$ in the sequence where $j < i$. This ordering ensures that the spatial context for every dependent object is fully defined prior to its placement.

#### 3.1.2. PROBABILISTIC SPATIAL MODELING

Following the sequential decomposition defined above, our objective is to model the conditional probability $P(\mathbf{B}_{dep,i}|\mathcal{A}_{dep,i}, \mathcal{O}_{sup,i}, \{\mathcal{O}_{fnc,i}\}_{opt})$ for each relational tuple $\mathcal{T}_i$. This distribution captures the spatial rules of the dependent object relative to the geometric and spatial features of its anchors.

Given the inherent multi-modality of object placement, where a single functional or support context might allow for several valid positions (e.g., a chair positioned at different orientations around a desk), we adopt a Mixture of Logistics (MoL) to represent the density function, which is similar to previous works (van den Oord et al., 2016; Salimans et al., 2017; Paschalidou et al., 2021). The model outputs a set of parameters $\Theta = \{\pi_k, \mu_k, s_k\}_{k=1}^K$ for $K$ mixture components. The probability density for the bounding box parameters $\mathbf{B}_{dep} \in \mathbb{R}^{12}$ is formulated as a weighted sum of $K$ logistic components:

$$P(\mathbf{B}_{dep}|\Theta) = \sum_{k=1}^K \pi_k \prod_{d=1}^{12} L(B_{dep,d}|\mu_{k,d}, s_{k,d}), \quad (2)$$

where $\pi_k$ represents the mixing coefficient, $\mu_k \in \mathbb{R}^{12}$ the mean vector, $s_k \in \mathbb{R}^{12}$ the scale parameter for the $k$-th component, and $L(\cdot \mid \mu, s)$ denotes the logistic distribution with location parameter $\mu$ and scale parameter $s$. By sampling from this learned multi-modal distribution, our framework can generate diverse yet physically and semantically consistent object placements within the scene sequence.

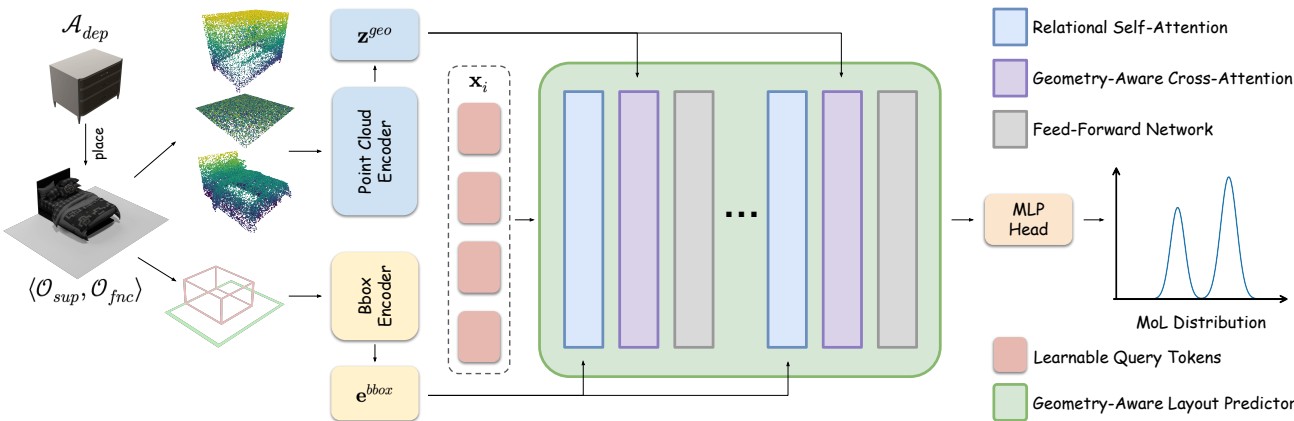

*Figure 2.* **Architecture of Pair2Scene Model.** The pipeline processes the dependent asset $\mathcal{A}_{dep}$ and anchor objects (support $\mathcal{O}_{sup}$ and functional $\mathcal{O}_{fnc}$) to predict the spatial distribution of the dependent object. Point cloud encoders extract latent geometric features $\mathbf{z}^{geo}$, while the bounding box encoder transforms spatial configurations into embeddings $\mathbf{e}^{bbox}$. These are processed through cascaded Transformer-like blocks consisting of Relational Self-Attention, which captures global relations using learnable query tokens $\mathbf{x}_m$, and Geometry-Aware Cross-Attention for fine-grained geometric reasoning. Finally, the MLP head outputs the parameters for a MoL distribution, providing a multi-modal probabilistic framework for object placement.

## 3.2. Learning Object Relations

Taking the dependent asset and the anchor assets alongside their layout as input, our model predicts the parameters of a Mixture of Logistics distribution as the probability density function for the dependent object's bounding box.

### 3.2.1. GEOMETRIC FEATURE ENCODING

Accurate spatial distribution modeling requires a fine-grained understanding of object geometry. Relying solely on semantic categories is insufficient for two primary reasons: first, support surfaces are often non-planar or do not coincide with the top face of a bounding box; second, it is inherently difficult to normalize the initial orientations across diverse objects.

To this end, we employ Point-MAE (Pang et al., 2022) as the geometric encoder, pretrained on the 3D asset library aggregated from the datasets in this work, to extract geometric features. For a given object, the encoder processes its point cloud to generate geometric features $\mathbf{z}_m^{geo} = \text{Encoder}(\mathcal{A}_m)$. By integrating these features, our model perceives the physical affordances and orientation of the objects, enabling the model to predict precise placements.

### 3.2.2. GEOMETRY-AWARE LAYOUT PREDICTOR

Geometry-Aware Layout Predictor, as illustrated in Fig. 2, fuses learnable latent queries with dense geometric features through a sequence of Transformer-like blocks (Vaswani et al., 2017). Each object $m \in \{dep, sup, fnc\}$ within a relational tuple is initially represented by a learnable query token $\mathbf{x}_m \in \mathbb{R}^d$, and we denote the set of all such tokens as $\mathbf{X} = \{\mathbf{x}_m\}_m$. For anchors, we incorporate their known spatial configurations by encoding the bounding box pa-

rameters $\mathbf{B}_m$ via an MLP to produce a spatial positional embedding $\mathbf{e}_m^{bbox} = \text{MLP}_{pos}(\mathbf{B}_m) \in \mathbb{R}^d$ (collectively denoted as $\mathbf{E}^{bbox}$). The core of the architecture consists of $N$ cascaded Transformer-like blocks.

**Relational Self-Attention.** Within each block, Relational Self-Attention layer captures the global relations between the dependent object and its anchors. By integrating the spatial embedding $\mathbf{e}^{bbox}$ into the key and value for anchors, the model enables the dependent to attend to the spatial presence of its anchors:

$$\mathbf{X} = \text{SelfAttn}(\mathbf{X}, \mathbf{X} + \mathbf{E}^{bbox}, \mathbf{X} + \mathbf{E}^{bbox}). \quad (3)$$

**Geometry-Aware Cross-Attention.** To capture geometric information, Geometry-Aware Cross-Attention layer attends to $\mathbf{z}_m^{geo}$ extracted from Point-MAE. Each token $\mathbf{x}_m$ performs the cross-attention operation exclusively on its own corresponding geometric features:

$$\mathbf{x}_m = \text{CrossAttn}(\mathbf{x}_m, \mathbf{z}_m^{geo}, \mathbf{z}_m^{geo}). \quad (4)$$

The tokens are then processed by a standard Feed-Forward Network (FFN) with residual connections and layer normalization to update the hidden states $\mathbf{X}$.

Finally, the refined latent representation of the dependent object $\mathbf{x}_{dep}$ is mapped through an output MLP head to predict the parameters $\hat{\Theta}$ of the MoL distribution:

$$\hat{\Theta} = \text{MLP}_{out}(\mathbf{x}_{dep}). \quad (5)$$

This distribution provides a multi-modal probabilistic framework for sampling the final predicted bounding box $\mathbf{B}_{dep}$.

### 3.2.3. TRAINING OBJECTIVE

Our model is trained to maximize the log-likelihood of the ground-truth bounding box parameters $\mathbf{B}_{dep}$ under the

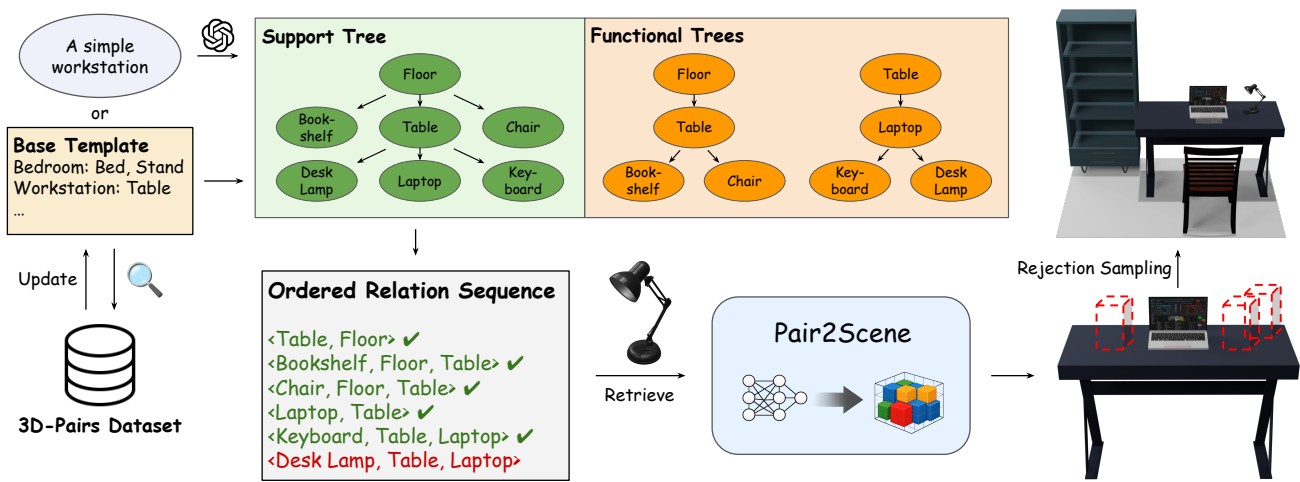

*Figure 3.* **Scene Assembly Pipeline.** The framework parses text or room type into support and functional Trees, serialized as an ordered relation sequence. The Pair2Scene model then predicts spatial distributions based on these relations. Finally, rejection sampling resolves global collisions to produce the final 3D scene.

predicted MoL distribution. Specifically, we minimize the Negative Log-Likelihood (NLL) loss function. For a target bounding box $\mathbf{B}_{dep} \in \mathbb{R}^{12}$, the NLL loss $\mathcal{L}_{nll}$ is formulated as:

$$\mathcal{L}_{nll} = -\log \sum_{k=1}^{K} \hat{\pi}_k \prod_{d=1}^{12} L(B_{dep,d}|\hat{\mu}_{k,d}, \hat{s}_{k,d}). \quad (6)$$

To prevent the model from collapsing and to encourage exploration of the multi-modal distribution space, we introduce an entropy regularization term $\mathcal{L}_{ent}$. This term rewards higher entropy in the mixing coefficients $\pi$:

$$\mathcal{L}_{ent} = \sum_{k=1}^{K} \hat{\pi}_k \log \hat{\pi}_k. \quad (7)$$

The total training objective $\mathcal{L}_{total}$ is defined as a weighted combination of the NLL loss and the entropy regularization term, governed by the hyperparameter $\lambda$:

$$\mathcal{L}_{total} = \mathcal{L}_{nll} + \lambda \mathcal{L}_{ent}. \quad (8)$$

### 3.3. Procedural Scene Assembly

To obtain a global scene from learned local rules, we represent the scene structure as a hierarchy of relations, which are subsequently serialized into an ordered sequence of relational tuples $\mathcal{S} = \{\mathcal{T}_1, \mathcal{T}_2, \ldots, \mathcal{T}_N\}$. By procedurally querying Pair2Scene model and employing rejection sampling, we resolve global spatial conflicts while maintaining physical plausibility (detailed in Alg. 2 and Fig. 3).

#### 3.3.1. RELATIONAL HIERARCHIES

To derive the generative sequence $\mathcal{S}$, we represent the global scene structure through two interconnected hierarchical structures that define the order of placement.

**Support Tree ($\mathbb{T}_s$):** This tree represents the global grounding hierarchy. The root node represents the floor, and each directed edge $(\mathcal{O}_i, \mathcal{O}_j)$ indicates that $\mathcal{O}_i$ serves as the support anchor for $\mathcal{O}_j$.

**Functional Tree ($\mathbb{T}_f$):** For every non-leaf node in $\mathbb{T}_s$, we define a functional tree. This structure captures the semantic dependencies of all objects sharing the same supporting surface. In this hierarchy, an edge indicates that the parent serves as the functional anchor for its child, while the root of $\mathbb{T}_f$ remains the common support anchor defined in $\mathbb{T}_s$.

To generate these trees, we offer two options: statistical synthesis, which utilizes occurrence frequencies and co-occurrence probabilities derived from our curated dataset to procedurally expand nodes (Alg. 1), and LLM-guided generation, which leverages the structural reasoning capabilities of LLM to design complex, commonsense-compliant scene hierarchies from text descriptions (see Appendix G).

#### 3.3.2. ORDERED RELATION SERIALIZATION

To transform these hierarchies into a generative sequence, we perform a hybrid traversal. We first traverse the Support Tree $\mathbb{T}_s$ using Breadth-First Search (BFS) to ensure that supporting surfaces (e.g., a table) are placed before the objects they support (e.g., a laptop). For each node encountered, we then perform a Depth-First Search (DFS) on its associated Functional Tree $\mathbb{T}_f$. This process yields a sequence of tuples $\mathcal{S}$, ensuring that every anchor is instantiated before it is used to condition the placement of a dependent object.

#### 3.3.3. COLLISION-AWARE REJECTION SAMPLING

While the layout predictor effectively captures local conditional densities, it lacks global awareness of objects outside the relational tuple. To account for global occupancy and

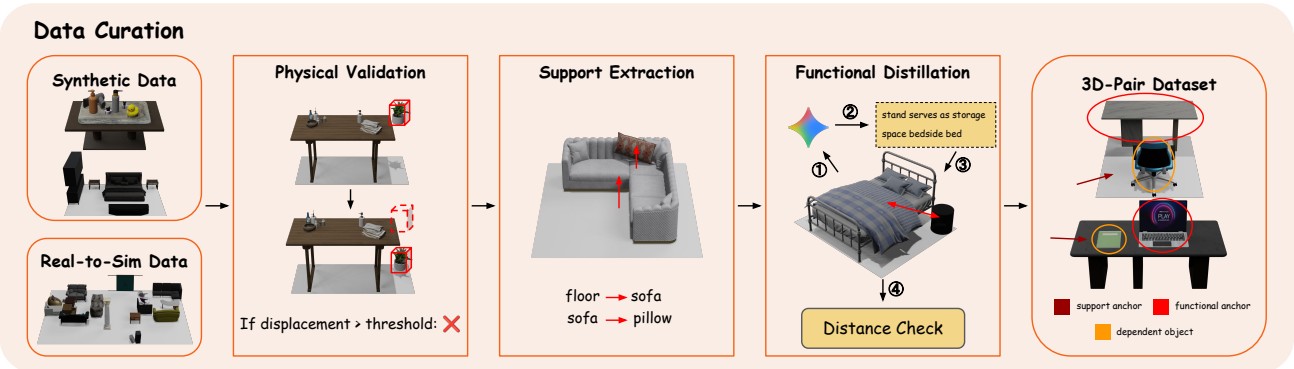

*Figure 4.* **Data Curation Pipeline.** Our data curation workflow involves Physical Validation to filter unstable objects, Support Extraction to identify support dependencies, and LLM-Driven Functional Distillation to extract functional relations. This process results in the 3D-Pair Dataset, which captures fine-grained object interactions.

prevent inter-object collisions, we define the target distribution $p_{\text{global}}(x)$ as a truncated version of the model's predicted distribution $p_{\text{local}}(x)$:

$$p_{\text{global}}(x) = \begin{cases} \frac{p_{\text{local}}(x)}{Z}, & x \in \mathcal{F} \\ 0, & x \notin \mathcal{F}, \end{cases} \quad (9)$$

where $\mathcal{F}$ denotes the set of physically feasible configurations (non-colliding states) within the current scene, and $Z = \int_{\mathcal{F}} p_{\text{global}}(x)dx$ is the normalization constant.

We approximate sampling from $p_{\text{global}}(x)$ via rejection sampling: we draw candidates from the predicted MoL distribution and discard those that result in collisions with placed assets or scene boundaries. Following a successful candidate, a brief gravity simulation is applied to refine placement. This mechanism effectively transforms local conditional predictions into a global scene configuration.

### 3.4. Data Curation

To support the learning of local rules, we develop a data curation pipeline that transforms global scene layouts from diverse datasets into the relational tuples $\mathcal{T}_i$ defined in Sec. 3.1. To construct a high-quality dataset of these local rules, we extract and refine data from 3D-Front (Fu et al., 2021), MesaTask (Hao et al., 2025), and the Real-to-Sim subset of InternScenes (Zhong et al., 2025). As shown in Fig. 4, our pipeline consists of three critical stages:

**Physical Validation and Filtering.** Given the inherent noise in large-scale scene datasets, particularly those generated through automated real-to-sim pipelines, we subject all scenes to a rigid-body physics simulation. By applying gravity and resolving initial collisions, we filter out unstable entities and optimize the final layout. Objects that exhibit significant displacement during simulation are discarded, ensuring that the layout is anchored in physical reality.

**Heuristic Support Extraction.** We identify $R_s$ pairs using a geometry-based heuristic. For any two proximal objects,

we evaluate whether the lower object's bounding box provides a stable base or whether it contains the upper object's bounding box. A support relation is established if the vertical proximity and horizontal containment exceed a defined threshold. Furthermore, to preserve the consistent distribution of large furniture in 3D-Front and mitigate issues arising from the noisy nature and inconsistent scaling of the InternScenes Real-to-Sim subset, we exclude data samples where the floor serves as the only anchor.

**LLM-Driven Functional Distillation.** For objects co-located on a shared surface, we leverage the commonsense knowledge and reasoning abilities of LLM (Google, 2025) to discern functional relations. The LLM identifies potential $R_f$ pairs and suggests a proximity factor $k$, which represents the expected interaction range. We validate these pairs by expanding the anchor object's bounding box by $k_f$. A functional relation is only archived if the dependent object's centroid remains within this expanded volume. This approach combines the reasoning of LLM with strict geometric verification to produce clean and rich relation pairs.

## 4. Experiments

### 4.1. Experimental Setup

**Evaluation Setup.** To assess the performance, we define two evaluation settings. The first is the **3D-Front only setting**, which evaluates the model's ability to fit the distribution of a dataset and generalize beyond this distribution. In this setting, we train and evaluate exclusively on the curated 3D-Front dataset and compare against learning-based baselines including ATISS (Paschalidou et al., 2021), DiffuScene (Tang et al., 2024), LayoutVLM (Sun et al., 2025a), and FactoredScenes (Hsu et al., 2025). We provide two variants of our model: *Ours-Fit*, which constrains object counts to match the dataset average to show fitting capability, and *Ours-Beyond*, which allows for unconstrained expansion to test the generation of scenes with complexity exceeding the

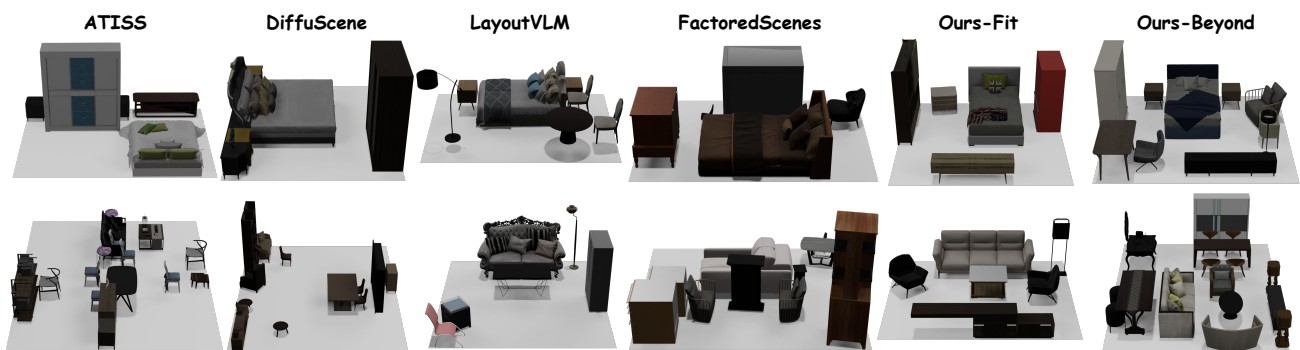

*Figure 5.* Qualitative comparison under the 3D-Front only setting.

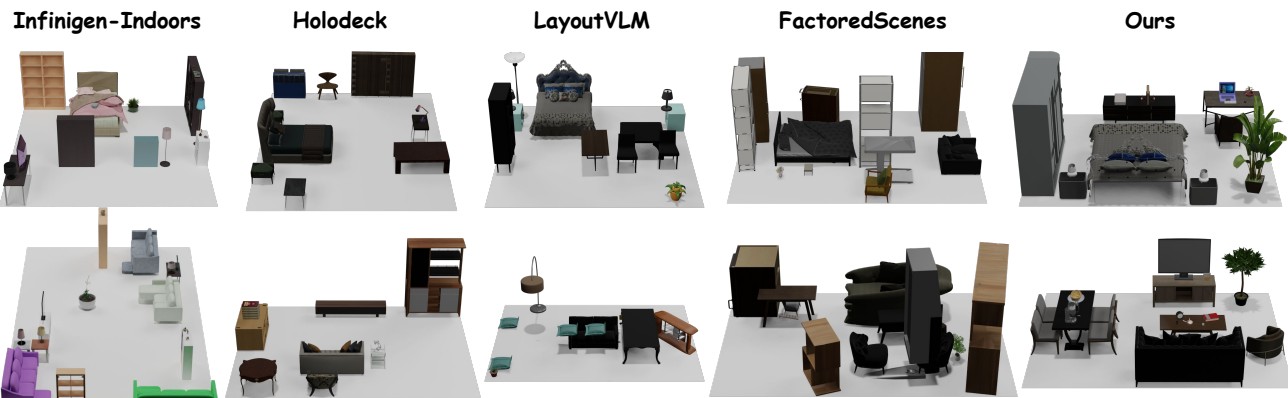

*Figure 6.* Qualitative comparison under the multi-source setting.

*Table 1.* Comparison of FID and KID on the 3D-Front only setting.

| Method | FID ↓ | KID ($\times 1e^{-3}$) ↓ | Objects |
|---|---|---|---|
| ATISS | $71.24 \pm 1.55$ | $42.18 \pm 1.42$ | 7.65 |
| DiffuScene | $67.45 \pm 0.92$ | $31.72 \pm 1.08$ | 6.75 |
| LayoutVLM | $120.87 \pm 2.05$ | $138.54 \pm 3.12$ | - |
| FactoredScenes | $104.12 \pm 2.78$ | $129.45 \pm 2.64$ | 8.53 |
| Ours-Fit | $65.92 \pm 1.14$ | $22.14 \pm 0.85$ | 6.98 |
| Ours-Beyond | $75.88 \pm 1.22$ | $69.05 \pm 1.58$ | 14.15 |

*Table 2.* User study results on the 3D-Front only setting.

| Method | SA ↑ | PP ↑ | SC ↑ | MQ ↑ | CFS ↑ |
|---|---|---|---|---|---|
| ATISS | 3.14 | 3.27 | 2.27 | 3.21 | 1.21 |
| DiffuScene | 4.45 | 4.14 | 2.86 | 4.30 | 2.05 |
| FactoredScenes | 1.73 | 2.27 | 2.82 | 2.00 | 0.94 |
| LayoutVLM | 2.27 | 2.14 | 3.91 | 2.21 | 1.44 |
| Ours-Fit | 4.18 | 4.18 | 3.91 | 4.18 | 2.73 |
| Ours-Beyond | 5.23 | 5.00 | 5.23 | 5.12 | 4.46 |

training distribution. The second is the **multi-source setting**, which assesses the capacity for high-complexity scene generation. We train on full 3D-Pairs (see Appendix B) and compare Pair2Scene against procedural and LLM-based frameworks, including Infinigen-Indoors (Raistrick et al., 2024), Holodeck (Yang et al., 2024b), LayoutVLM, and FactoredScenes, which have demonstrated capabilities in generating detailed scenes.

**Metrics.** Regarding the evaluation metrics, for the 3D-Front only setting, we quantify generation quality using Fréchet Inception Distance (FID) (Heusel et al., 2017) and Kernel Inception Distance (KID) (Bińkowski et al., 2018) calculated from Bird's-Eye View (BEV) renders of 200 generated scenes following the protocol of DiffuScene (Tang et al., 2024). These metrics measure the similarity and distribution alignment between generated and real scenes.

Furthermore, we conduct a user study on both settings with 22 participants to evaluate the scenes across three qualitative criteria: (1) *Semantic Alignment* (SA), assessing if the layout aligns with human commonsense; (2) *Physical Plausibility* (PP), evaluating the absence of artifacts like floating objects or collisions; and (3) *Scene Complexity* (SC), measuring the richness and functional density of the scene. Participants rank the methods from best to worst. We report a score for each metric, calculated as $\sum (Frequency \times Weight)/TotalSamples$, where the weight for rank $r$ is defined as $N - r + 1$. Finally, we report two holistic measures: *Mean Quality* (MQ), the average of SA and PP; and the *Contextual Fidelity Score* (CFS), which weights MQ by the relative Scene Complexity to reflect overall scene quality in context.

*Table 3.* User study results on the multi-source setting.

| Method | SA ↑ | PP ↑ | SC ↑ | MQ ↑ | CFS ↑ |
|---|---|---|---|---|---|
| FactoredScenes | 1.50 | 2.05 | 1.91 | 1.78 | 0.68 |
| Holodeck | 3.36 | 3.59 | 2.09 | 3.48 | 1.45 |
| Infinigen-Indoors | 2.50 | 2.73 | 3.09 | 2.62 | 1.62 |
| LayoutVLM | 3.09 | 2.32 | 3.18 | 2.71 | 1.72 |
| Ours | 4.55 | 4.32 | 4.73 | 4.44 | 4.20 |

*Table 4.* Ablation study results on the 3D-Front only setting.

| Variant | FID ↓ | KID ($\times 1e^{-3}$) ↓ |
|---|---|---|
| w/o relation | 92.34 | 82.74 |
| w/o pretrain | 81.14 | 73.91 |
| w/o rejection | 78.26 | 68.56 |
| Class-only | 68.45 | 35.23 |
| w/o gravity | 66.82 | 28.48 |
| Full | 65.15 | 22.32 |

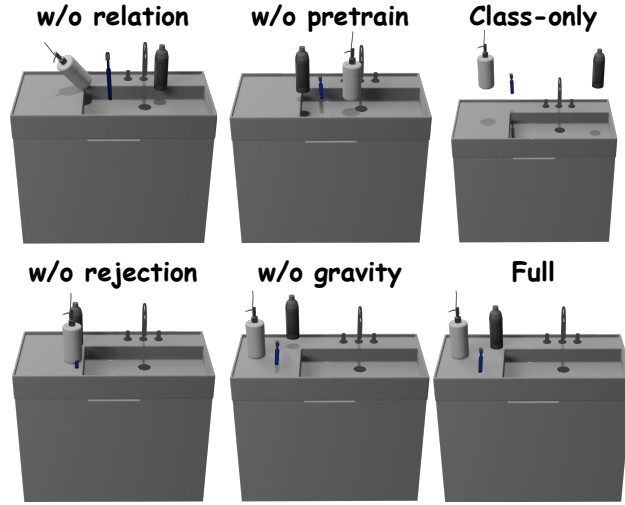

*Figure 7.* Qualitative ablation study on the multi-source setting.

## 4.2. Fitting and Extrapolation on 3D-Front

The 3D-Front only setting evaluates the model's capacity to capture the distribution of the dataset while testing its ability to surpass the complexity limits of the dataset. As shown in Table 1, *Ours-Fit* variant achieves the best performance, outperforming established learning-based methods. This indicates that our local rule modeling more accurately represents the spatial distributions of indoor scenes compared to global layout predictors, such as ATISS and DiffuScene. These quantitative gains are further corroborated by the user study results in Table 2. While baselines remain confined to the limited object density of the training set, *Ours-Beyond* variant leverages the procedural nature of the framework to expand the scene, achieving a superior Scene Complexity score. As evidenced by the high Semantic Alignment and Physical Plausibility scores, this increase in density does not compromise layout logic. As shown in Fig. 1 and 5, while other methods struggle to scale, our method generates spatially coherent and semantically plausible extensions that go beyond the original dataset's distribution, reflected in our dominant Contextual Fidelity Score.

## 4.3. Hierarchical Scene Generation on 3D-Pairs

Training on the full 3D-Pairs enables Pair2Scene to generate complex scenes that integrate large furniture with fine-grained arrangements. As reflected in Table 3, Fig. 1 (Bottom) and Fig. 6, our approach consistently surpasses all baselines. While LayoutVLM and FactoredScenes often suffer from floating artifacts or collisions as object density increases, Pair2Scene maintains superior semantic alignment and physical plausibility. Furthermore, unlike procedural and LLM-based baselines such as Infinigen-Indoors and Holodeck, which often produce sparse or semantically disconnected layouts, our method generates rich environments

where objects are logically grounded. This is evidenced by our significantly higher scores in Scene Complexity and Semantic Alignment, where human evaluators find our layouts more functionally dense and consistent with commonsense. Overall, our framework excels in balancing intricate detail with overall scene coherence, achieving the highest Contextual Fidelity Score among all methods.

## 4.4. Ablation Study

We ablate key designs of Pair2Scene on both settings to verify their contributions as shown in Fig. 7 and Table 4. Specifically, we evaluate: (1) *w/o relation*, using global coordinate regression instead of local rules; (2) *w/o pretrain*, training the Point-MAE from scratch; (3) *Class-only*, replacing point cloud inputs with category embeddings; (4) *w/o rejection*, disabling the rejection sampling mechanism and using the initial raw samples; and (5) *w/o gravity*, removing the physics-based gravity simulation.

**Impact of Pretraining.** The gap between *w/o pretrain* and our full model highlights that geometric features from our integrated asset library are crucial for the layout predictor to accurately perceive object scale and shape.

**Relational vs. Global Modeling.** The *w/o relation* variant shows the sharpest performance drop, proving that regressing global layout underperforms compared to our local rules modeling, which effectively captures spatial relations.

**Geometric Priors vs. Semantic Labels.** Compared to *Class-only*, although our model shows marginal gains in FID/KID, metrics mainly sensitive to top-down distributions, the visual results in Fig. 7 reveal underlying issues regarding physical plausibility. By incorporating geometric priors, our model ensures more stable interactions.

**Effectiveness of Rejection Sampling.** The performance

degradation in the *w/o rejection* variant (Table 4) demonstrates the necessity of our sampling strategy. Without it, the model often produces severe object interpenetrations and fails to align with the global scene distribution, as it lacks the awareness of the current scene configurations.

**Role of Gravity Simulation.** While the *w/o gravity* variant shows relatively stable performance in FID and KID, visual inspection in Fig. 7 reveals its limitations. Without gravity simulation, the model struggles with imprecise contact surfaces, often resulting in floating objects or unstable support relations. Our full model ensures more realistic physical affordance.

## 5. Conclusion

This paper presents Pair2Scene, a relational framework addressing scalability and data scarcity in 3D indoor scene generation. By decomposing global synthesis into localized procedural rules, our approach captures complex spatial distributions more effectively than existing global methods. Leveraging 3D-Pairs and a Geometry-Aware Layout Predictor, Pair2Scene successfully bridges asset geometry with spatial logic. Experiments demonstrate that our framework excels at both fitting dataset distributions and generating high-density scenes exceeding the complexity observed in the training data. This shift toward relational modeling establishes a robust and scalable foundation for learning-based scene generation.

**Limitations.** While effective, Pair2Scene has limitations. The focus on spatial relations does not guarantee stylistic consistency during object retrieval. Additionally, the model struggles with corner cases, such as objects governed by multiple anchors or uncommon relations (e.g., hanging objects). Future work will address these challenges.

## Acknowledgements

This research is supported in part by the HKU Startup Fund, the Institute of Data Science, and HUAWEI's AI Hundred Schools Program. Some experiments were carried out using the Ascend AI technology stack.

## Impact Statement

This paper presents a relational framework designed to advance the field of machine learning by improving the scalability and spatial reasoning of 3D indoor scene generation. These advances have direct positive implications for the game industry, film production, and virtual reality by automating the creation of detailed virtual worlds. Furthermore, our framework contributes significantly to Embodied AI and robotics by providing diverse, high-quality simulated environments for training agents to navigate and interact with human spaces safely. There are many other potential societal consequences of our work, none of which we feel must be specifically highlighted here.

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

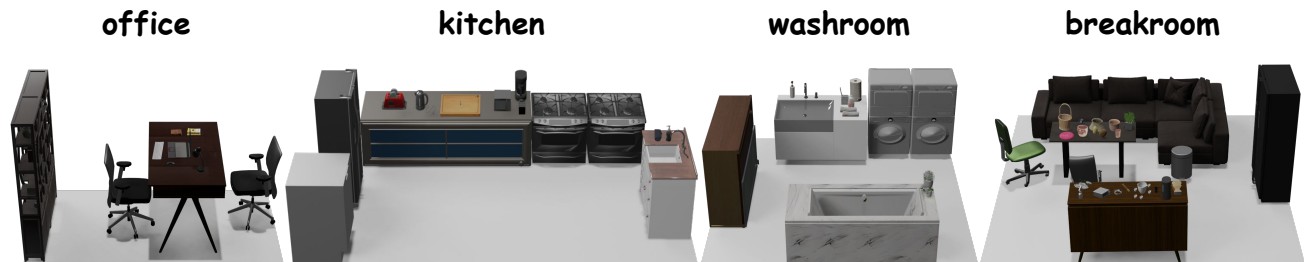

*Figure 8.* More visualization results on new scene types.

*Table 5.* Comparison of Inference Time (Seconds per Scene).

| Method | 3D-Front only | Multi-source |
|---|---|---|
| ATISS (Paschalidou et al., 2021) | 9.45s | - |
| DiffuScene (Tang et al., 2024) | 28.16s | - |
| Holodeck (Yang et al., 2024b) | - | 210.23s |
| Infinigen-Indoors (Raistrick et al., 2024) | - | 686.44s |
| LayoutVLM (Sun et al., 2025a) | 247.92s | 373.69s |
| FactoredScenes (Hsu et al., 2025) | 85.26s | 85.26s |
| Ours-Fit | 26.89s | - |
| Ours-Beyond / Ours (Full) | 49.28s | 69.19s |

## A. Implementation Details

For the implementation, we utilize Point-MAE (Pang et al., 2022) as the point cloud encoder, pretrained on our integrated 3D asset library. During training, the input assets are randomly sampled to 8,192 points. The latent dimension $d$ is set to 512, and the layout predictor consists of 7 Transformer-like blocks. Each object in a relational tuple is represented by 4 learnable query tokens, and the MoL distribution is configured with $K = 4$ mixture components. The model was trained for 120 epochs using the AdamW optimizer with a learning rate of $1 \times 10^{-4}$ on 8 NVIDIA A100 (40G) GPUs, with a total batch size of 32. For training stability, we normalize the support anchor's bounding box to a canonical cube and predict the dependent's distribution within the local coordinate. In addition, the support trees and functional trees of Fig. 1, 5, and 6 are generated using the statistical synthesis approach. The additional results 8 are produced via LLM-guided generation using simple text prompts, such as "a kitchen scene." Regarding the metrics in Table 1, we generated 20 test scenes for each of the bedroom and living room categories for evaluation.

## B. Datasets.

Our experimental evaluation is designed to validate the efficacy of the Pair2Scene framework across diverse scene generation tasks. Following the data curation pipeline described in Sec. 3.4, we extract a total of 138,429 relational tuples. This large-scale dataset aggregates 23,023 relations from 3D-Front (Fu et al., 2021) (comprising 4,530 bedrooms and 987 living rooms) for large furniture layouts, 56,719 relations from MesaTask (Hao et al., 2025) for high-density tabletop arrangements, and 58,687 relations from Real-to-Sim subset of InternScenes (Zhong et al., 2025) to enhance generalization to real scenes. Detailed information regarding the data distributions can be found in Fig. 10 through 18.

## C. Inference Time Analysis

In this section, we provide a detailed analysis of the computational efficiency of Pair2Scene compared to existing baselines. We measure the average time required to generate a complete scene across both the 3D-Front only and multi-source settings. As summarized in Table 5, Pair2Scene exhibits competitive inference efficiency. In the 3D-Front only setting, ATISS remains faster due to its lightweight autoregressive design, while our *Ours-Fit* variant achieves comparable inference time to diffusion-based methods such as DiffuScene, despite providing stronger relational reasoning capabilities. Although the inference time for our *Ours-Beyond* and multi-source configurations increases due to the generation of higher scene complexity and a larger number of objects, it remains substantially more efficient than LLM-based or procedural frameworks. For instance, in the multi-source setting, Pair2Scene generates highly dense scenes in 69.19s, whereas LayoutVLM and

*Table 6.* Comparison of Accuracy Rate and Resampling per Object.

| Method | Acc. Rate ↑ | Resample/Obj. ↓ |
|---|---|---|
| Ours-Fit | 75.63% | 0.322 |
| Ours-Beyond | 57.12% | 0.751 |

*Table 7.* Quantitative validation of LLM-based extraction and ablation study of the $k$-check mechanism. We report mean and standard deviation across 30 random scenes per source.

| Datasource | Precision ↑ | Prec. (w/o $k$) | Recall ↑ | Rec. (w/o $k$) | F1 ↑ | F1 (w/o $k$) |
|---|---|---|---|---|---|---|
| **3D-Front** | $0.924 \pm 0.065$ | $0.909 \pm 0.062$ | $0.921 \pm 0.059$ | $0.927 \pm 0.070$ | 0.922 | 0.918 |
| **MesaTask** | $0.846 \pm 0.041$ | $0.777 \pm 0.058$ | $0.933 \pm 0.085$ | $0.967 \pm 0.073$ | 0.887 | 0.862 |
| **InternScenes** | $0.800 \pm 0.095$ | $0.763 \pm 0.073$ | $0.807 \pm 0.110$ | $0.819 \pm 0.067$ | 0.803 | 0.790 |

Holodeck require 373.69s and 210.23s respectively. This efficiency stems from our hierarchical formulation, which allows for localized distribution sampling rather than expensive global scene refinement or long-sequence autoregressive tokens.

## D. Generalization to New Scene Types

To demonstrate the generalization capability of our approach, Fig. 8 presents additional visualization results on novel scene categories. Despite the domain shift, our model is able to synthesize semantically coherent and physically plausible layouts by leveraging learned relational rules, indicating that our framework can extrapolate beyond the original dataset distribution for new but structurally related scene types.

## E. Rejection Sampling Efficiency

To quantify the overhead of our collision-aware rejection sampling, we measure the acceptance rate and the average resamples per object. As shown in Table 6, even in the high-density Ours-Beyond setting, the acceptance rate remains above 50% with less than one resample required per object on average. This efficiency demonstrates that our local rule modeling provides a high-quality initial proposal that is already largely consistent with physical constraints, rather than relying on brute-force sampling to find valid placements.

## F. LLM-based Extraction Validation

To validate the reliability of LLM-based spatial relation extraction, we evaluate its performance against human expert annotations on a subset of 30 randomly sampled scenes from each datasource. As summarized in Table 7, our method achieves high F1-scores across diverse environments, demonstrating strong robustness in structural understanding.While LLMs excel at capturing functional context, they are prone to "semantic hallucinations", erroneously predicting relations between semantically related but spatially distant objects. To mitigate this, we introduce the $k$-check mechanism, which enforces distance constraints tailored to specific relation types. Our ablation study confirms that removing the proximity factor $k$ consistently leads to a drop in precision due to these hallucinations. The $k$-check is therefore essential for ensuring spatial precision with negligible impact on recall.

## G. LLM Prompt for Scene Hierarchy Generation

This section provides the detailed system prompt and task instructions used to guide LLM in generating structured scene hierarchies. The model is tasked with interpreting textual descriptions to produce physically plausible support relations ($\mathbb{T}_s$) and semantic functional dependencies ($\mathbb{T}_f$).

```
System Role: You are an expert spatial reasoning assistant specializing in
    architectural scene synthesis and hierarchical relational modeling.
```

```
Task:
Convert a user-provided textual description into a structured scene hierarchy
    consisting of a Support Tree ($\mathbb{T}_s$) and multiple Functional Trees ($\
    mathbb{T}_f$).

Definitions:
- Support Tree ($\mathbb{T}_s$): A global grounding hierarchy.
  * Root: Always the "Floor".
  * Edges: A directed edge ($O_i, O_j$) means $O_i$ physically supports $O_j$ (e.g.,
      Table supports Lamp).
- Functional Tree ($\mathbb{T}_f$): A semantic dependency hierarchy for objects
    sharing the same support surface.
  * Root: The common support anchor from $\mathbb{T}_s$.
  * Edges: A directed edge ($O_i, O_j$) means $O_i$ is the functional anchor for $O_j$
      (e.g., Chair is placed relative to a Desk; Plate is placed relative to a
      Placemat).

Output Format:
You must output a valid JSON object with the following structure:
{
  "support_tree": [
    {"parent": "Floor", "child": "Object_A"},
    {"parent": "Object_A", "child": "Object_B"}
  ],
  "functional_trees": [
    {
      "support_anchor": "Object_A",
      "edges": [
        {"parent": "Object_A", "child": "Object_B"},
        {"parent": "Object_B", "child": "Object_C"}
      ]
    }
  ]
}

User Input:
"{{SCENE_DESCRIPTION}}"

Instructions:
1. Identify all objects mentioned or implied by the scene.
2. Construct the Support Tree ensuring every object is grounded back to the "Floor".
3. For every node in the Support Tree that has children, construct a Functional Tree
    to define the semantic layout logic between those children.
4. Ensure physical plausibility and commonsense spatial relationships.
```

## H. User Study

Fig. 9 shows the interface used in our user study. Participants are presented with multiple scenes and are asked to rank the scenes according to specific evaluation criteria. For each question, users inspect rendered scenes and drag-and-sort the options from best to worst based on a designated metric, following the provided positive and negative examples. This interface enables consistent and intuitive collection of human preference rankings across different evaluation dimensions. In addition, the CFS in Table 2 and 3 is specifically calculated as:

$$\text{CFS} = \frac{\text{MQ} \times \text{SC}}{\text{Total Samples}} \tag{10}$$

## I. Floating Rate

We investigate the vertical stability of generated scenes using the floating rate. As shown in Table 8, without the gravity simulation, the model suffers from a significant floating rate. In contrast, our full model effectively eliminates these artifacts,

*1. In this question, you will see a set of room designs containing **only large furniture** (e.g., beds, cabinets, etc.). Please rank the following layouts from best to worst based **only** on the metric of **[Semantic Alignment]**.

**Evaluation Criteria:** Does the furniture placement align with human living habits?
Positive Examples: Chairs facing tables; headboards positioned against walls; TVs facing sofas.
Negative Examples: Cabinet doors blocked by other furniture and unable to open; furniture facing the wrong direction.

**Action:** Please place the layout with the best **[Semantic Alignment]** at the top. Please select all options and **click the top right corner to drag to** sort.

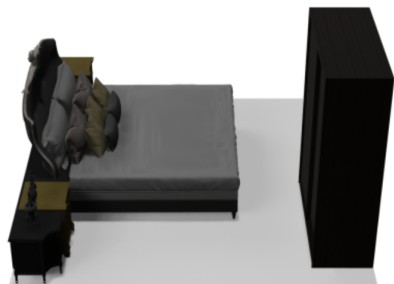

B

*Figure 9.* Screenshot of the user study interface.

*Table 8.* Ablation analysis of the floating rate.

| Variant | Floating Rate ↓ |
|---|---|
| w/o gravity | $0.27 \pm 0.12$ |
| Full (Ours) | $0$ |

achieving a zero floating rate.

## J. Collision Rate

We evaluate the physical plausibility of the generated scenes using the collision rate, defined as the ratio of colliding objects to the total number of objects within a scene. Table 9 presents a comprehensive comparison against baselines and a detailed ablation study of our proposed components. Compared to other baselines, we achieve the best performance thanks to the learned local spatial distributions combined with rejection sampling. On the other hand, our full model significantly outperforms existing global layout predictors. The ablation results further highlight the importance of our architectural choices. Notably, the w/o relation variant lacks sampling rejection due to compatibility constraints. The substantial performance gap between the full model and the variant without rejection sampling demonstrates that our rejection mechanism effectively minimizes physical intersections.

*Table 9.* Collision rate results: comparison with baselines (left) and ablation study (right).

| Comparison with Baselines | | Ablation Study | |
|---|---|---|---|
| Method | Collision Rate ↓ | Variant | Collision Rate ↓ |
| ATISS | $0.31 \pm 0.14$ | w/o relation | $0.33 \pm 0.18$ |
| DiffuScene | $0.24 \pm 0.11$ | w/o pretrain | $0.07 \pm 0.06$ |
| LayoutVLM | $0.27 \pm 0.06$ | w/o rejection | $0.25 \pm 0.08$ |
| FactoredScenes | $0.25 \pm 0.09$ | Class-only | $0.03 \pm 0.04$ |
| Ours-Fit | $0.02 \pm 0.05$ | w/o gravity | $0.02 \pm 0.05$ |
| Ours-Beyond | $0.05 \pm 0.03$ | Full | $0.02 \pm 0.05$ |

---

**Algorithm 1** Generate Scene Hierarchies

---

1: **Input:** Scene Type $\mathcal{P}$, Statistical Dependencies $\mathcal{D} = \{sup_{dep}, func_{dep}\}$, Max Objects $N_{max}$, Probability Coefficient $k$
2: **Output:** Support Tree $\mathbb{T}_s$, Functional Trees $\{\mathbb{T}_f\}$
3: $\{\mathbb{T}_s, \{\mathbb{T}_f\}\} \leftarrow$ InitializeBaseTemplate($\mathcal{P}$) {Initial scene setup}
4: $Counts \leftarrow$ InitializeInstanceCounts($\mathbb{T}_s$)
5: $AddedCount \leftarrow 0$
6: $Queue \leftarrow []$ {Stores tuples of (LeafNodeID, AnchorNodeID)}
7: {1. Identify Extendable Anchors and Leaf Nodes}
8: **for** each $AnchorID, \mathbb{T}_{f,Anchor} \in \{\mathbb{T}_f\}$ **do**
9:    $L \leftarrow$ GetBaseLabel($\mathbb{T}_s[AnchorID]$)
10:    $Leaves \leftarrow \{n \mid n \in \text{Nodes}(\mathbb{T}_{f,Anchor}), \text{Children}(n) = \emptyset\}$
11:    **for** each $n \in Leaves$ **do**
12:      $Queue$.append($(n, AnchorID)$)
13:    **end for**
14: **end for**
15: {2. Recursive Expansion based on Dependency Probabilities}
16: **while** $Queue$ is not empty **and** $AddedCount < N_{max}$ **do**
17:    $(u_{leaf}, u_{anchor}) \leftarrow Queue$.pop(0)
18:    $L_{anc}, L_{leaf} \leftarrow$ MapToCategory($\mathbb{T}_s[u_{anchor}], \mathbb{T}_s[u_{leaf}]$)
19:    **if** $L_{leaf} \in sup_{dep}[L_{anc}]$ **then**
20:      $TotalFreq \leftarrow sup_{dep}[L_{anc}][L_{leaf}]$
21:      $Candidates \leftarrow func_{dep}[L_{anc}][L_{leaf}]$
22:      **for** each $(L_{cand}, Freq_{cand}) \in Candidates$ **do**
23:        $P \leftarrow k * Freq_{cand}/TotalFreq$
24:        **if** random$(0, 1) < P$ **and** $AddedCount < N_{max}$ **then**
25:          {Instantiate and link new node}
26:          $L_{new} \leftarrow$ GenerateUniqueLabel($L_{cand}, Counts$)
27:          $\mathbb{T}_s$.AddNode($L_{new}, \text{parent} = u_{anchor}$)
28:          $\mathbb{T}_{f,u_{anchor}}$.AddNode($L_{new}, \text{parent} = u_{leaf}$)
29:          $Queue$.append($(u_{anchor})$)
30:          $AddedCount \leftarrow AddedCount + 1$
31:        **end if**
32:      **end for**
33:    **end if**
34: **end while**
35: **return** $\mathbb{T}_s, \{\mathbb{T}_f\}$

---

---

**Algorithm 2** Procedural Scene Assembly

---

1: **Input:** 3D Asset Library $\mathcal{V}$, Text Prompt/Scene Type $\mathcal{P}$, Layout Predictor $\mathcal{M}$
2: **Output:** Global 3D Scene $\mathcal{G}$
3: $\mathcal{G} \leftarrow \emptyset$
4: $Z_{floor} \leftarrow$ Instantiate floor as root of $\mathbb{T}_s$
5: $\mathcal{G} \leftarrow \mathcal{G} \cup \{Z_{floor}\}$
6: {1. Joint Hierarchical Structuring}
7: $\{\mathbb{T}_s, \{\mathbb{T}_f\}\} \leftarrow$ GenerateSceneHierarchies($\mathcal{P}$) {Support and Functional trees}
8: {2. Ordered Relation Serialization}
9: $\mathcal{S} \leftarrow []$
10: $Nodes \leftarrow$ BFS($\mathbb{T}_s$)
11: **for** each $\mathcal{O}_i \in Nodes$ **do**
12:     $\mathcal{S}_{sub} \leftarrow$ DFS($\mathbb{T}_{f,i}$) {Yields tuples of (anchor(s), dependent)}
13:     $\mathcal{S}$.append($\mathcal{S}_{sub}$)
14: **end for**
15: {3. Sequential Placement with Rejection Sampling}
16: **for** each relational tuple $\mathcal{T}_k = (\text{anchors}, \text{dependent}) \in \mathcal{S}$ **do**
17:     $\mathcal{A}_{dep} \leftarrow$ RetrieveAsset(dependent, $\mathcal{V}$) {Retrieve asset from library}
18:     Placed $\leftarrow$ False
19:     **while** not Placed **do**
20:         $x \sim p_{local}(x \mid \text{anchors}, \mathcal{A}_{dep}, \mathcal{M})$ {Sample conditioned on anchors and asset}
21:         **if** $x \in \mathcal{F}$ **and** $x$ within boundaries **then**
22:             $x^* \leftarrow$ GravityRefinement($x$)
23:             $\mathcal{G} \leftarrow \mathcal{G} \cup \{x^*\}$
24:             Placed $\leftarrow$ True
25:         **else**
26:             Reject candidate $x$
27:         **end if**
28:     **end while**
29: **end for**
30: **return** $\mathcal{G}$

---

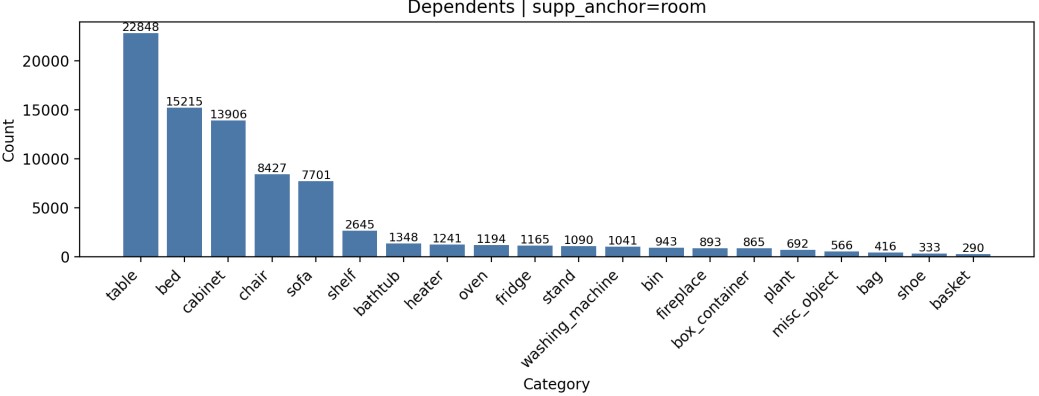

*Figure 10.* Distribution of dependents for floor as support anchor.

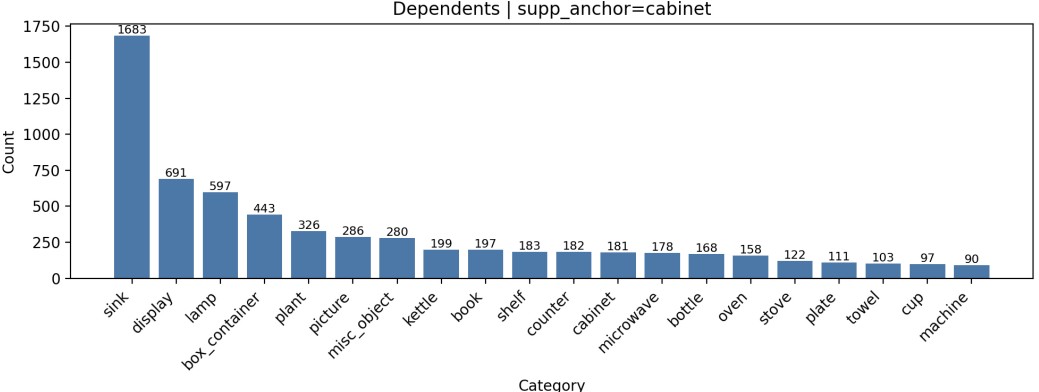

*Figure 11.* Distribution of dependents for cabinet as support anchor.

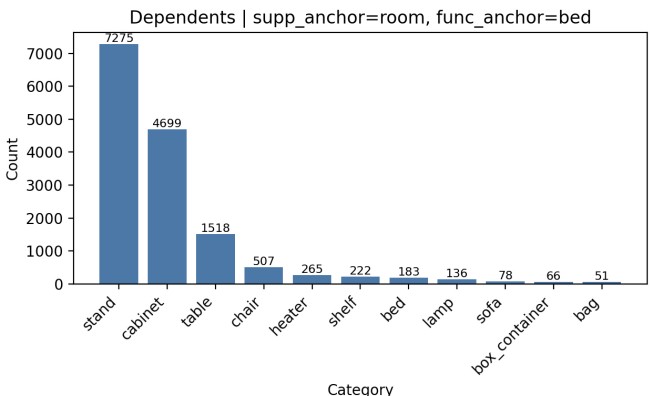

*Figure 12.* Distribution of dependents for floor as support anchor and bed as functional anchor.

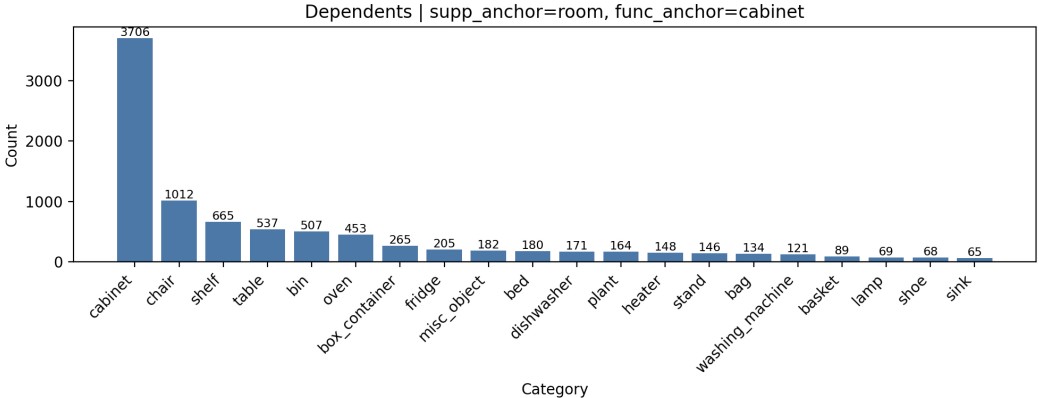

*Figure 13.* Distribution of dependents for floor as support anchor and cabinet as functional anchor.

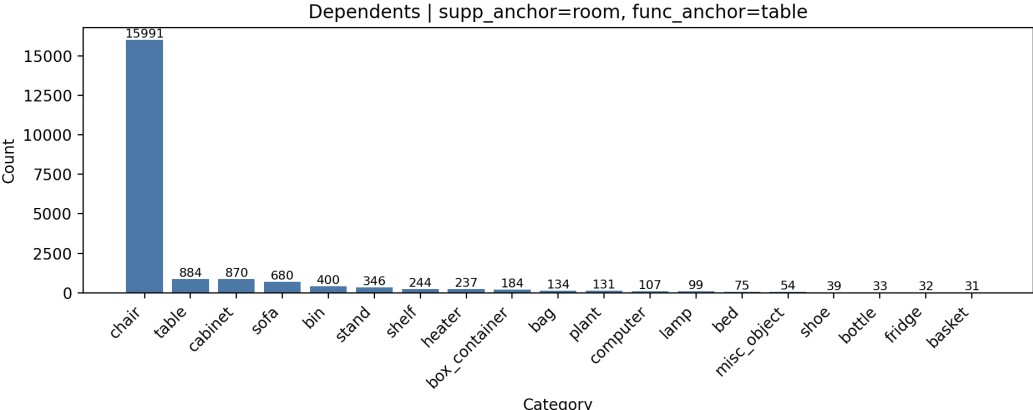

*Figure 14.* Distribution of dependents for floor as support anchor and table as functional anchor.

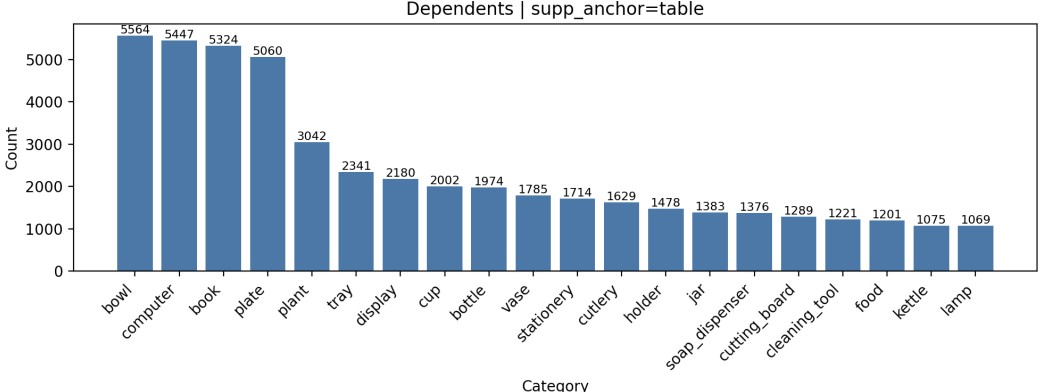

*Figure 15.* Distribution of dependents for table as support anchor.

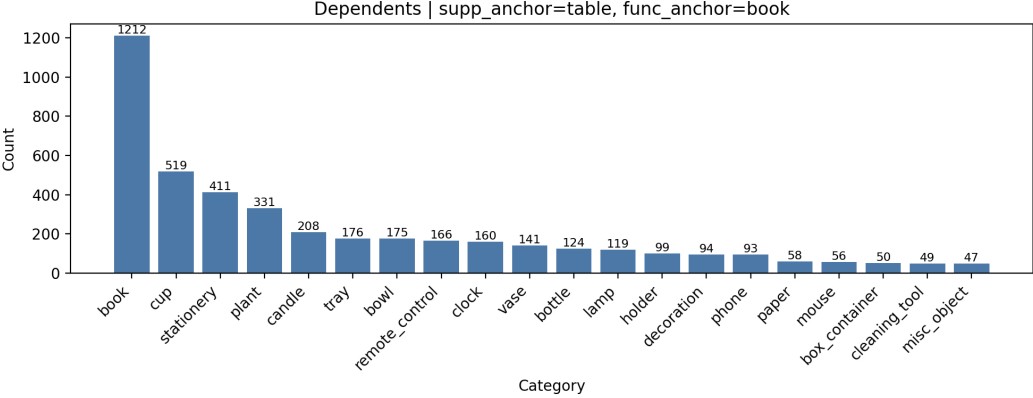

*Figure 16.* Distribution of dependents for table as support anchor and book as functional anchor.

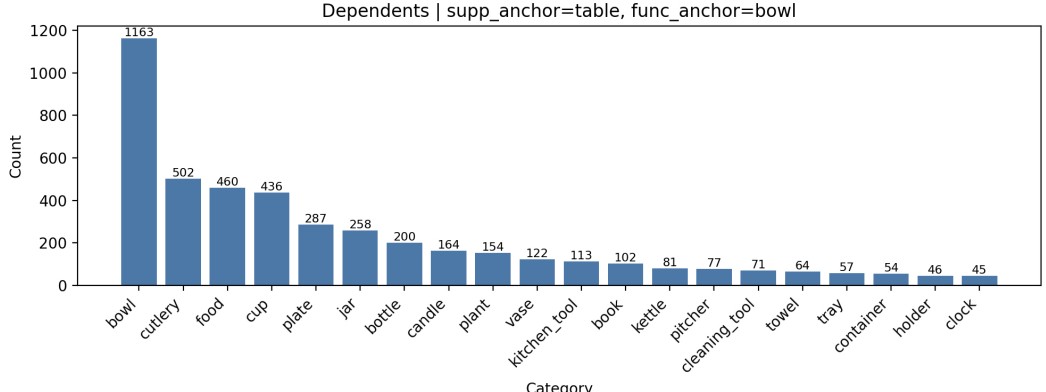

*Figure 17.* Distribution of dependents for table as support anchor and bowl as functional anchor.

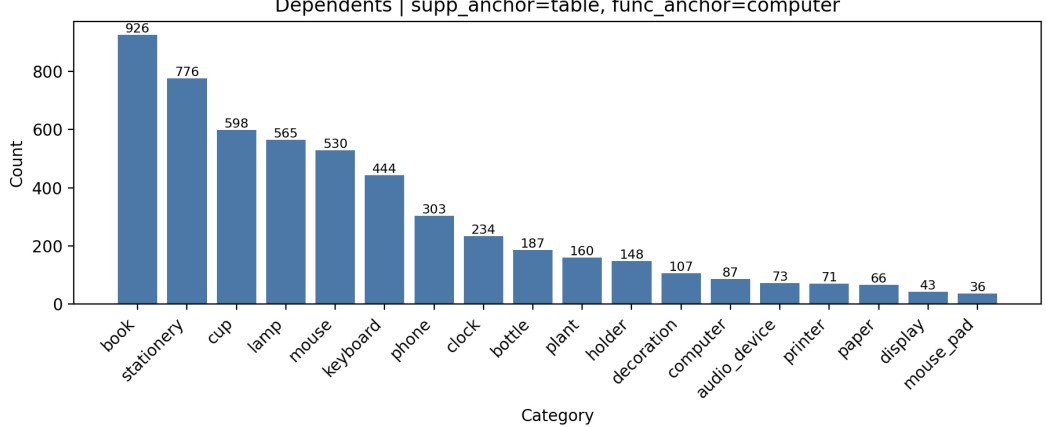

*Figure 18.* Distribution of dependents for table as support anchor and computer as functional anchor.

