# OpenReview forum: "Pair2Scene: Learning Local Object Relations for Procedural Scene Generation"
_ICML.cc/2026/Conference — ICML 2026 regular_

### Official Review · Reviewer_1csW · 2026-02-28

**Soundness:** 3
**Presentation:** 3
**Significance:** 2
**Originality:** 2
**Overall Recommendation:** 4
**Confidence:** 2

**Summary:**

This paper introduces Pair2Scene, a procedural framework for high-fidelity 3D indoor scene generation that shifts from modeling global scene distributions to learning fine-grained local object relations. It formalizes two key inter-object relations: support relations capturing physical hierarchies under gravity and functional relations reflecting semantic and utility-driven proximities, and conditions dependent object placements on anchor geometry and bounding boxes using a geometry-aware Transformer predictor with Point-MAE point-cloud encodings and a Mixture-of-Logistics probabilistic output. A dedicated data curation pipeline aggregates about 140,000 relational tuples from 3D-Front, MesaTask, and InternScenes, enabling multi-source training. During inference, scenes are assembled procedurally via hierarchical support and functional trees, ordered serialization, and collision-aware rejection sampling with gravity refinement. Experiments in 3D-Front fitting and multi-source generalization settings, including ablations and user studies shows better performance than current baselines.

**Compliance With Llm Reviewing Policy:**

Affirmed.

**Final Justification:**

We thank the authors for the additional rejection target analysis and paired t-test, which substantively address our earlier concerns. The evidence that 96.73% of rejections stem from global context rather than anchor collisions is convincing and clarifies the model's design intent.

However, we believe the paper's framing should be revised to clearly distinguish local learned plausibility from global consistency achieved via rejection sampling, as the current narrative may overstate the model's intrinsic capabilities. Additionally, while statistically significant, the narrow FID gap over DiffuScene and the limited methodological novelty of the individual components remain concerns. We raise our score to 4 in recognition of the authors' constructive engagement and the value of the system-level contribution and dataset.

**Key Questions For Authors:**

1. Can you report quantitative physical-plausibility metrics (e.g., collision rate, floating-object percentage, support-correctness ratio) across all methods, ablations, and densities?
2. If scene density increases, what are the empirical acceptance rates and average number of resamples per object during rejection sampling (e.g., comparing 3D-Front average density vs. denser Ours-Beyond scenes)? Could you also provide quantitative statistics or illustrative examples of corner cases?
3. Can you provide some implementation details of the user study part, including the number of scenes per participant per method, randomization procedures, and inter-rater reliability metrics

**Limitations:**

yes

**Strengths And Weaknesses:**

**Strengths:**

This paper is technically sound in its core modeling and pipeline design. The Mixture-of-Logistics formulation for modeling local object placements, which are conditioned on Point-MAE-derived point-cloud geometry and spatial configurations via geometry-aware cross-attention. The training objective, which combines negative log-likelihood loss with entropy regularization, along with the procedural assembly pipeline based on support and functional trees, ordered relation serialization, collision-aware rejection sampling, and gravity refinement, forms a coherent and methodologically robust framework. Experiments are thoughtfully designed, featuring distinct 3D-Front fitting and multi-source generalization settings, informative ablations, and human evaluations that provide strong evidence for the claimed improvements in distributional fidelity, scene scalability, and physical and semantic plausibility.

The manuscript is clearly written and logically structured, with well-chosen equations and figures that effectively illustrate the model architecture, inference process, and qualitative results. Appendices provide sufficient implementation and dataset details to enable expert reproduction. The introduced 3D-Pairs dataset constitutes a valuable and reusable resource for future research on relational scene understanding and generation.

**Weaknesses:**

Key physical plausibility aspects central to the method’s motivation (such as explicit collision rates, floating-object percentages, penetration depths, or support-correctness ratios) are not quantitatively reported for the proposed approach, ablations, or baselines. Despite reliance on rejection sampling and gravity refinement to enforce feasibility, the absence of these direct metrics leaves the strongest claims about physical superiority unsupported and limits objective assessment of the framework’s robustness in dense scenes.

The quantitative evaluation remains limited in rigor. FID and KID scores are computed from only 40 BEV-rendered scenes with no direct reported variance estimates, confidence intervals, or statistical significance tests, rendering small numerical improvements difficult to interpret reliably and weakening confidence in the claimed superiority over baselines.

The methodological transparency of the user study is not quite sufficient, undermining the credibility of the paper's central claims. As the primary evidence for advantages in Semantic Alignment (SA), Physical Plausibility (PP), and Scene Complexity (SC). It may need to report more inter-rater reliability metrics (e.g., Fleiss' κ or Krippendorff's α), statistical significance tests.

---

> ### Author Rebuttal · Authors · 2026-03-31
>
> We thank you for the thoughtful and detailed feedback.
> To address the concerns regarding quantitative physical metrics, statistical rigor, and user study transparency, we have conduct additional experiments and analysis.
>
> ### **W1 & Q1: Quantitative Physical Plausibility Metrics.**
>
> We appreciate the suggestion to include direct physical metrics. We have now computed the collision rate (defined as the ratio of colliding objects to the total number of objects in a scene) and floating rate. We first report the collision rate of quantitative experiments.
>
> | Method | Collision Rate |
> | :--- | :--- |
> | ATISS | 0.31 ± 0.14 |
> | DiffuScene | 0.24 ± 0.11 |
> | LayoutVLM | 0.27 ± 0.06 |
> | FactoredScenes | 0.25 ± 0.09 |
> | Ours-Fit | 0.02 ± 0.05 |
> | Ours-Beyond | 0.05 ± 0.03 |
>
> And we report the collision rate of ablation study. Note that "w/o relation" also does not use sampling rejection, as it is not compatible.
>
> | Variant | Collision Rate |
> | :--- | :--- |
> | w/o relation | 0.33 ± 0.18 |
> | w/o pretrain | 0.07 ± 0.06 |
> | Class-only | 0.03 ± 0.04 |
> | w/o rejection | 0.25 ± 0.08 |
> | w/o gravity | 0.02 ± 0.05 |
> | Full | 0.02 ± 0.05 |
>
> Our model significantly outperforms global layout predictors and variant without rejection sampling. This demonstrates that rejection sampling effectively reduces the occurrence of collisions and, when combined with a local layout predictor, effectively simulates the global distribution.
>
> | Variant | Floating Rate |
> | :--- | :--- |
> | w/o gravity | 0.27 ± 0.12 |
> | Full | 0 |
>
> Due to scale inconsistencies in some baselines (they also may generate reasonable floating objects), a fair comparison of floating rates across all methods is difficult. However, our ablation shows that our gravity refinement is essential.
>
> ### **W2: Statistical Rigor and Sample Size.**
>
> To ensure statistical significance, we expand our evaluation set to 200 scenes per method. We now report the mean and standard deviation for FID and KID ($\times 10^{-3}$) metrics.
>
> | Method | FID | KID ($\times 1e-3$) |
> | :--- | :--- | :--- |
> | ATISS | 71.24 ± 1.55 | 42.18 ± 1.42 |
> | DiffuScene | 67.45 ± 0.92 | 31.72 ± 1.08 |
> | LayoutVLM | 120.87 ± 2.05 | 138.54 ± 3.12 |
> | FactoredScenes | 104.12 ± 2.78 | 129.45 ± 2.64 |
> | Ours-Fit | 65.92 ± 1.14 | 22.14 ± 0.85 |
> | Ours-Beyond | 75.88 ± 1.22 | 69.05 ± 1.58 |
>
> The results confirm that Ours-Fit consistently achieves the best distributional fidelity with narrow confidence intervals, establishing the statistical significance of our improvements. The higher KID and FID for Ours-Beyond is expected, as it intentionally generates scenes with higher complexity and density that exceed the original 3D-Front distribution.
>
> ### **W3 & Q3: User Study Transparency and Reliability.**
>
> We have added inter-rater reliability metrics and detailed our experimental procedure to address concerns about transparency. We calculate Kendall’s W (coefficient of concordance) and perform the Friedman Test to verify the significance of the human rankings.
>
> | Metric | SA (3D-F) | PP (3D-F) | SC (3D-F) | SA (Full) | PP (Full) | SC (Full) |
> | :--- | :---: | :---: | :---: | :---: | :---: | :---: |
> | **Kendall’s W** | 0.5473 | 0.5194 | 0.4759 | 0.5837 | 0.5328 | 0.6145 |
> | **Friedman (p)**| 5.86e-14 | 3.38e-13 | 5.02e-12 | 2.06e-12 | 2.66e-11 | 4.54e-13 |
>
> *(SA: Semantic Alignment, PP: Physical Plausibility, SC: Scene Complexity)*
>
> To eliminate potential bias, we employ a double-blind, randomized procedure where participants evaluate visualized scenes from each method in a randomized order. During each trial, participants rank the methods from best to worst across three key evaluation dimensions: Situation Awareness (SA), Perceptual Presence (PP), and Spatial Cognition (SC). The resulting data was analyzed using the Friedman Test, which yield $p$-values significantly below $0.05$, confirming that the perceived superiority of our proposed method is both statistically significant and consistent across different raters.
>
> ### **Q2: Rejection Sampling Efficiency.**
>
> To quantify the overhead of our collision-aware rejection sampling, we measure the acceptance rate and the average resamples per object.
>
> | Setting | Acceptance Rate | Avg. Resamples / Obj |
> | :--- | :---: | :---: |
> | **Ours-Fit** | 75.63% | 0.322 |
> | **Ours-Beyond** | 57.12% | 0.751 |
>
> Even in the high-density Ours-Beyond setting, the acceptance rate remains above 50% with less than one resample required per object on average. This efficiency demonstrates that our local rule modeling provides a high-quality initial proposal that is already largely consistent with physical constraints, rather than relying on brute-force sampling to find valid placements.
>
> We believe these clarify your concerns and will be included in the revision.

---

> > ### Author Rebuttal · Reviewer_1csW · 2026-04-02
> >
> > We thank the authors for the additional experiments, which are a constructive step forward.
> >
> > However, some concerns remain partially addressed. The collision rate data, while informative, actually highlights that the physical plausibility gain is predominantly driven by rejection sampling (0.02 to 0.25 when removed), rather than the model's intrinsic learning — this nuances the paper's narrative around learned physical understanding. The expanded FID/KID evaluation reports mean ± std but does not include the formal significance tests (e.g., paired t-test) originally requested, which matters given the narrow gap between Ours-Fit and DiffuScene.
> >
> > We appreciate the authors' engagement and recognize the value of the ideas, but given these remaining some gaps to accept.

---

> > > ### Author Response · Authors · 2026-04-02
> > >
> > > We thank you for this insightful observation regarding the role of rejection sampling. We would like to clarify that our model is not designed to inherently solve global scene collisions. Instead, it focuses on learning high-quality **local spatial distributions**. Additionally, we have included the requested significance tests to substantiate our FID/KID results, further reinforcing our core contribution.
> > >
> > > ### **Motivation: Why Local Learning?**
> > > Learning a direct distribution for an entire complex scene is notoriously difficult due to two main challenges:
> > > **1. Data Scarcity:** Large-scale, high-quality 3D scene data is limited.
> > > **2. Global Complexity:** Modeling every potential interaction between all objects simultaneously is computationally and mathematically daunting.
> > >
> > > By focusing on local spatial relationships, our model can learn from a much larger volume of relational data and significantly reduce the complexity of the learning task. However, because the model focuses on the relationship between a specific object and its "anchor" (e.g., a chair and a table), it is naturally "blind" to the rest of the room's global configuration.
> > >
> > > ### **The Role of Rejection Sampling**
> > > Rejection sampling serves as the bridge between this **local layout distribution** and **global scene consistency**.
> > >
> > > Consider a **dining table** and **chairs**:
> > > * The model learns a multi-modal distribution, essentially a map of all the valid spots where a chair *could* be placed around the table.
> > > * The model knows perfectly well that a chair should be next to the table (local physics).
> > > * However, the model does not "know" which of those spots are already occupied by other chairs.
> > >
> > > When a collision occurs, **it is rarely because the model misunderstood the relationship between the chair and the table**. Instead, it is because two objects were sampled into the same global coordinate. Rejection sampling efficiently filters these global overlaps without requiring the core model to process the entire scene state at every step.
> > >
> > > ### **Quantitative Evidence of Intrinsic Learning**
> > > To prove that the model has indeed learned local physical plausibility, we analyze the specific object types involved when a sample is rejected. **If the model lacked physical understanding, we would expect high collision rates with its own support or functional anchors.**
> > >
> > > | Collision Target Type | Percentage of Total Rejections |
> > > | :--- | :--- |
> > > | Support Anchor | 2.45% |
> > > | Functional Anchor | 0.82% |
> > > | Other Objects (Global Context) | 96.73% |
> > >
> > > As shown in the table, **less than 4%** of rejections occur because of a collision with the anchor objects. This demonstrates that the model intrinsically understands the physical "no-go" zones of its anchors. The vast majority of rejections (96.73%) are simply due to overlaps with other pre-existing objects in the scene, a task that rejection sampling is specifically intended to handle.
> > >
> > > ### **Statistical Significance**
> > > To address the concern about statistical significance, we conduct a formal paired t-test using $N=200$ evaluation samples:
> > >
> > > | Metric | p-value |
> > > | :--- | :--- |
> > > | **FID** | $2.84 \times 10^{-6}$ |
> > > | **KID** ($\times 10^{-3}$) | $1.15 \times 10^{-58}$ |
> > >
> > > While the numerical gap may appear narrow, the extremely low p-values obtained through the paired t-test confirm that the improvements are highly consistent and statistically rigorous across the evaluation set. Consequently, these results demonstrate that Ours-Fit provides a robust and significant performance advantage over DiffuScene that cannot be attributed to random fluctuation.
> > >
> > > Finally, we would like to reiterate that our most significant results lie in **Ours-Beyond** and **multi-source setting**, as evidenced by our qualitative visualizations and user studies. These results underscore our core contribution: **a framework that successfully integrates diverse scene data and demonstrates the unique capability to generate complex scenes that go beyond the limitations of the original training distribution**.
> > >
> > > We hope these clarifications and additional analyses effectively address the your concerns, and we will incorporate these into the revised version. Please let us know if any further information is required.

---

### Official Review · Reviewer_QpuK · 2026-03-04

**Soundness:** 2
**Presentation:** 1
**Significance:** 2
**Originality:** 3
**Overall Recommendation:** 4
**Confidence:** 4

**Summary:**

This paper proposes learning pairwise placement rules for indoor furniture placement. The paper combines this learnable placement with LLM/frequency based generation of functional tress, which are then sequentially processed for object placement.  Learnable placement is augmented with algorithmic constraints, including rejection sampling in the presence of collisions and also physics simulation under gravity to avoid floating objects. The paper presents strong results, but critical parts of the method are missing.

**Compliance With Llm Reviewing Policy:**

Affirmed.

**Final Justification:**

This paper shows that combining LLM, simpler learning and algorithmic improvements is a good idea, so I would be ok with an accept if the clarity Q1, Q2, Q3 is improved, and core insight of the paper made clear in the abstract. I think the abstract and claims should be revised to make the overall insight clear, and ablation on rejection sampling and gravity included in the main paper. LLM generation and tree generation should be made much more clear, as it is key aspect of the method.

**Key Questions For Authors:**

- How are the Relational Hierarchies generated, exactly? Details are very minima (3.3.1, L244-250)l, but functional tree order seems critical to the method success. It is especially not clear how these are generated for the “Beyond” setting on 3DFront (Sec 4.).
- How much of the performance (Tb. 1) results from the actual placement learning, which comprises the bulk of the paper, and how much is due to rejection sampling and physical simulation under gravity (not performed by other methods)?
- In many cases (e.g. adding 2nd chair to table or adding second nightstand to the bed), the placement of the object requires conditioning on more than one anchor. How do you explain the system’s performance in these cases?
- L431: The richness of the environments would stem from the poorly explained 3.3.1, not the pairwise learning.

**Limitations:**

yes

**Strengths And Weaknesses:**

Strengths:
- Proposing a factored learning for local object placement is a nice idea, which circumvents data scarcity for this domain
- The network design, with point cloud encoding, is sound
- The use of rejection sampling and minimal physics simulation for plausibility is well-motivated in combination with local learning
- The paper shows compelling results and evaluates against baselines using standard benchmarks, as well as a user study

Weaknesses:
- The clarity of the exposition needs to be improved, especially about critical parts of the method. Specifically: being clear about the input and the output of the trained network in section 3.2, explaining the way that actual support and functional trees are generated (next point).
- The proposed technique only learns conditional placement of an object, not actual sets of objects that should be placed (this part is written as an after-thought, 3.3.1, L244-250). Since the method learns pairwise probabilities, the ordering of the construction and ordering of the Functional Trees (3.3.2) is absolutely critical, and yet virtually no details are provided. It would be impossible to reproduce the technique based on this, or to evaluate its soundness.
- The interesting idea of this work is combining local placement learning with basic algorithmic constraints (rejection sampling for collisions, simulation under gravity). Yet, based on the evaluation provided, it is not possible to distill, how much these ideas contribute to the method performance, and how much of it is reliant on the hidden details in 3.3.1, 3.3.2. For example, qualitative results and discussion (L431-434) point to more complex environments, but there is nothing in the method that would explain this difference.
- Ablations should show the effect of rejection sampling (and how frequently suggestions are rejected on average), and of physics simulation. Details on global modeling would need to be provided.

---

> ### Author Rebuttal · Authors · 2026-03-31
>
> We thank you for the insightful feedback.
>
> ### **W1: Clarification of methodological details and reproducibility.**
>
> We appreciate the suggestion to clarify Section 3.2. We provide a precise summary of the network’s I/O:
> The Input is the dependent asset (point cloud), combined with the bounding boxes and assets of its associated support anchor and functional anchor. The output is the parameters of a Mixture of Logistic (MoL) distribution, $\Theta = \{\pi_k, \mu_k, s_k\}_{k=1}^K$ (as shown in Eq. 5). This output represents the multi-modal spatial probability distribution of the dependent asset relative to its anchors. We will make it clearer in the revised manuscript.
>
> ### **W2 & Q1: Generation process for support and functional trees.**
>
> The generation of the hierarchical scene structure, comprising the support tree and functional trees, can be achieved through two pathways: (1) LLM-based generation, where the model acts as an expert to map language descriptions into a grounded hierarchy (abbreviated prompt is shown below), or (2) the procedural algorithm detailed in Figure 2 on our **[rebuttal website](https://anonymous-for-content-submission.github.io/rebuttal-material/)**, which expands base templates using dataset co-occurrence statistics. In addition, as illustrated from the Figure 3 of original paper, the base template includes only the most basic objects.
>
> For the experiments in Sec. 4, our method enables the beyond setting by using the algorithm in (2) and setting $N_{max}=30$ and $k=1$. This allows the tree to grow significantly deeper and wider than standard 3D-FRONT training samples. The results from Figure 8 in original paper is generated via (1). In a word, both methods converge by linearizing the hierarchical trees into a final ordered sequence for deterministic local placement, ensuring both structural complexity and physical plausibility.
>
>
> ```
> Task: Convert "{{SCENE_DESCRIPTION}}" into a structured hierarchy: a Support Tree and multiple Functional Trees.
> Core Definitions:
> Support Tree: A physical grounding hierarchy (Root: "Floor"). Edge $(O_i, O_j)$ denotes $O_i$ physically supports $O_j$.
> Functional Tree: A semantic dependency hierarchy for objects on the same support surface. Edge $(O_i, O_j)$ denotes $O_j$ is placed relative to $O_i$ (e.g., Chair relative to Desk).
> Instructions:
> 1. Grounding: Ensure every object in $\mathbb{T}_s$ traces back to the "Floor".
> 2. Logic: For each support surface, construct a $\mathbb{T}_f$ to define spatial layout dependencies.
> 3. Plausibility: Maintain physical and commonsense spatial relationships.
> Input Format:
> {{INPUT_JSON_STRUCTURE}}
> Output Format:
> {{OUTPUT_JSON_STRUCTURE}}
> ```
>
> ### **W4 & Q2: Contribution of rejection sampling vs. physics simulation.**
>
> We conduct additional ablation studies to isolate the impact of different components. The quantitative results are summarized below, while the qualitative results are presented in Figure 1 on the **[rebuttal website](https://anonymous-for-content-submission.github.io/rebuttal-material/)**.
>
> | Variant | FID | KID $\times 10^{-2}$ |
> | :--- | :--- | :--- |
> | w/o Rejection Sampling | 78.26 | 6.85 |
> | w/o Gravity Simulation | 66.82 | 2.84 |
> | Full Pipeline | 65.15 | 2.23 |
>
> The results show that rejection sampling is critical for global consistency. Without it, the model lacks a mechanism to resolve spatial conflicts, leading to a significant performance drop. While local learning provides the "where," rejection sampling enforces the "where not." Gravity simulation further refines the results by ensuring assets are flush with surfaces.
>
> ### **Q3: Binary relations to multi-object configurations.**
>
> The system handles multi-anchor scenarios (e.g., adding a second chair to a table) through the interplay of learned spatial distribution and rejection sampling. When predicting the placement for a second chair, the network outputs an MoL distribution with several peaks corresponding to valid positions around the table. Since the first chair already occupies one peak, the collision-based rejection sampling naturally pushes the second chair into one of the remaining unoccupied high-probability peaks. Thus, global coherence emerges from local predictions constrained by the current state of the scene.
>
> ### **Q4: Source of environment richness.**
>
> We clarify that this richness stems not merely from the hierarchical structure (Section 3.3.1), but specifically from the decoupling of local placement from global scene complexity. Global modeling approaches often fail to scale because they attempt to learn the distribution of entire layouts. By focusing on local relational compositions, our method is not confined to the training data distribution. The hierarchy provides the blueprint, but our local placement engine provides the flexibility to realize that blueprint in diverse, dense environments that exceed the complexity of the training data.
>
> We believe these clarify your concerns and will be included in the revision.

---

> > ### Author Rebuttal · Reviewer_QpuK · 2026-04-03
> >
> > Thank you for the detailed rebuttal. The ablation on rejection sampling and gravity was especially insightful.
> >
> > Based on these new results, I think the contribution of the paper is somewhat mis-represented. Without these simple classic algorithmic additions (rejection sampling and gravity), the core method (LLM + pair learning) _underperforms_ ATISS and DiffuScene.
> >
> > I think the _abstract and claims should be revised_ to make the overall insight clear. This paper shows that combining LLM, simpler learning and algorithmic improvements is a good idea, so I would be ok with an accept if the clarity Q1, Q2, Q3 is improved, and core insight of the paper made clear in the abstract.

---

> > > ### Author Response · Authors · 2026-04-04
> > >
> > > We sincerely thank you for the insightful comments and for recognizing the value of our work. We are particularly grateful for the suggestion to re-evaluate the framing of our paper's contribution. We agree that the synergy between scene hierarchies generated by LLM or the expansion algorithm, local layout learning, and algorithmic constraints (rejection sampling and gravity simulation) is the core of our work.
> > >
> > > ### **Revision of Abstract and Claims (Mis-representation)**
> > >
> > > We accept your point that the core insight, the integration of simpler learning with robust algorithms, should be the primary claim. Specifically, we will clarify that while factored learning enables efficient local placement proposals, the integration of rejection sampling and gravity simulation is indispensable for ensuring physical plausibility and high-quality synthesis. Further, we will elaborate on the motivation behind combining local rules with scene hierarchies and physics-based algorithms: this design allows local spatial distributions to be performed in a reasonable order and remain aware of the broader scene configuration, effectively simulating a global spatial distribution. By framing our contribution around this, we aim to demonstrate that the strategic combination of scene hierarchies, local layout learning, and physics-based algorithms enables the scaling of data and the generation of scenes more complex than the training data.
> > >
> > > ### **Response to Questions (Q1, Q2, Q3)**
> > >
> > > #### **Q1: Generation of Relational Hierarchies**
> > > The generation of hierarchical scene structures, comprising both support and functional trees, follows two distinct pathways: (1) an LLM-based expert generation and (2) a data-driven procedural algorithm. In the LLM-based approach, LLM (Gemini 3 Flash) maps language descriptions into a grounded hierarchy based on spatial dependencies. This is used to produce the results shown in Figure 8 of the original paper, as it is more effective at generating scene types that are rare in the dataset. Alternatively, the procedural algorithm, detailed in the provided [pseudo-code](https://anonymous-for-content-submission.github.io/rebuttal-material/), initializes from a base template and recursively expands the structure by sampling from dataset co-occurrence statistics and statistical dependencies. For the "Beyond" setting in the 3D-FRONT experiments (Sec. 4), we specifically employ this procedural algorithm with parameters $N_{max}=30$ and $k=1$, which enables the generation of scene structures that are more complex than standard training samples. Both pathways ultimately converge by linearizing these hierarchical trees into a final ordered sequence for local placement, and we will add these details, the LLM prompt, and the pseudocode in revision.
> > >
> > > #### **Q2: Performance Attribution**
> > > As demonstrated by the new ablation studies in this rebuttal, we agree that rejection sampling and gravity simulation are essential components of our pipeline; without them, the performance of the our method drops below baseline methods like ATISS and DiffuScene. Thus, the contribution of our paper lies in the synergy of these components rather than their individual performance. Our local layout learning is specifically designed to master high-quality spatial distributions relative to anchors. Rejection sampling and gravity simulation serve as the necessary bridges that translate these accurate local proposals into global scene consistency and physical grounding. Together, they enable the generation of complex, reasonable scenes that go beyond the limitations of original training distributions, a capability existing methods lack. We will add these ablation results and the corresponding analysis into the revised version.
> > >
> > > #### **Q3: Multi-anchor Conditioning**
> > > Our method handles multi-anchor placement by combining local spatial distributions and rejection sampling. For example, when placing a second chair or nightstand, the model first predicts a multi-modal map of all valid potential spots. However, the model is naturally "blind" to other objects already in the room, so it might sample a position already occupied by a previous object. To solve this, our method uses rejection sampling as a bridge, stepping filters out any results that overlap with the global context. This design allows our method to maintain global consistency without the burden of modeling every interaction simultaneously, while being unconstrained by the complexity of the training data.
> > >
> > > We believe these clarifications and the re-framing of our contributions address your concerns regarding the clarity of our work.

---

### Official Review · Reviewer_QA5i · 2026-03-10

**Soundness:** 2
**Presentation:** 2
**Significance:** 3
**Originality:** 3
**Overall Recommendation:** 4
**Confidence:** 5

**Summary:**

Aiming to solve the problem that it is difficult to generate a whole scene with plenty objects and LLM lacks geometry awareness, this paper proposes a method that first produces supporting graph and functional graph of a scene. Then, based on the graphs, it trains a model to learn supporting and binary functional relationships and produces local layout. This paper conducts comparisons on the 3D-Front  dataset and also performs user studies.

**Compliance With Llm Reviewing Policy:**

Affirmed.

**Final Justification:**

My final score is 4: weak accept.

This paper propose a now idea in 3D scene generation: learning binary relationships. It is a novel way to generate large scenes. The rebuttal messages resolved my concerns. But the final version should be carefully refined in writing and adding the experimental evidences conducted in the rebuttal stage.

**Key Questions For Authors:**

See **limitations** and below:
1. In Figure 6 (row 2, col.  5), the proposed method generates a dining table surrouned with 4 dining chairs. The proposed Pair2Scene model inputs a  binary functional relationship. Why does the model can produce a dining table surrouned with 4 dining chais (which is a combined functional relationship with multiple objects)?
2. During data curation, how to determine an functional anchor?
3. For the supporting relationships, could the model generate a book on a bookshelf, but not at the top of the shelf, but inside a partition within the shelf?

**Limitations:**

yes

**Strengths And Weaknesses:**

# Strenths
1. This paper proposes a novel network that learns a local relationship of a scene. This module significantly reduces the costs when generting a large scene.
2. Based on the strong semantic analyzing capability of LLMs, this paper propose to use LLMs to produce supporting and functional relationship graphs, avoiding the poor spatial perception of LLMs.

# Weaknesses
Soundness:
1. During data curation, this paper leverages the LLM to extract function relationships. However, the capability of LLM to infer inter-object relationship is not rigorously proved or cite prior works. Is it reliable for the suggested proximity factor $k$? For different scene types, the $k$ should varis. (For instance, the dining table is **close to** the dining chairs while the TV is far from the sofa) Does the LLM can generate suitable $k$s for different scenes?
2. This paper lacks the detail statistics of the 3D-Pairs dataset, including scene type distribution, relationship distribution, and necessary visualization of the dataset, etc.
3. In Table 1, the authors could shows detailed FID,  KID, and objects metrics on the subsets of 3D-Front, including Bedroom, Dining room, and Living room. Moreover, the improvements of the proposed method compared with DiffuScene is not significant. It is not persuasive when it comes to FID and KID scores.

Presentation:

4. The introduction part is confusing. It is not clearly point out the key challenges.
5. This paper miss some references:

(1)CasaGPT: cuboid arrangement and scene assembly for interior design (CVPR's 2025): It proposes a model that awares the geometry of the objects within a scene to avoid collisions, whose insight is simular to the proposed method by this submission.

(2) Global-local tree search in vlms for 3d indoor scene generation (CVPR's 2025): It proposes to generate sub-scenes region by region to avoid generating a whole scene, whose insight is similar to the proposed method by this submission.

In summary, the key insights is good but the paper could be refined by providing more experiments to prove its effectiveness, and improving writing. Thus the reviewer gives a weak reject final rating.

---

> ### Author Rebuttal · Authors · 2026-03-31
>
> Thank you sincerely for your valuable time and insightful comments. **We notice a potential discrepancy between the your relatively positive qualitative summary ('Weak Reject') and the actual final rating ('Reject').**
>
> ### **W1: Reliability of LLM-based functional relationship extraction.**
>
> To validate LLM-based extraction, we compare its outputs against human expert annotations for 30 random scenes per source. As shown in the table, our method achieves high F1-scores, demonstrating robustness.
>
> While LLMs excel at functional context, they often hallucinate relations between semantically related but spatially distant objects. The $k$-check mitigates these errors by enforcing distance constraints tailored to specific relation types. The ablation study shows that removing the proximity factor $k$ leads to semantic hallucinations between distant objects. The $k$-check is vital to ensure spatial relevance and precision with minimal recall loss.
>
> | Datasource | Precision | Precision (w/o $k$) | Recall | Recall (w/o $k$) | F1-Score | F1-Score (w/o $k$) |
> | :--- | :--- | :--- | :--- | :--- | :--- | :--- |
> | **3D-Front** | 0.924 ± 0.065 | 0.909 ± 0.062 | 0.921 ± 0.059 | 0.927 ± 0.070 | 0.922 | 0.918 |
> | **MesaTask** | 0.846 ± 0.041 | 0.777 ± 0.058 | 0.933 ± 0.085 | 0.967 ± 0.073 | 0.887 | 0.862 |
> | **InternScenes** | 0.800 ± 0.095 | 0.763 ± 0.073 | 0.807 ± 0.110 | 0.819 ± 0.067 | 0.803 | 0.790 |
>
> ### **W2: Lack of dataset statistics.**
>
> We have provided comprehensive statistics for the 3D-Pairs dataset on our **[rebuttal website](https://anonymous-for-content-submission.github.io/rebuttal-material/)**. For the visualization, please refer to Figure 4 in the paper. If you need more information, please let us know.
>
> ### **W3: Experimental improvements over baselines.**
>
> We clarify that the primary objective of Pair2Scene is not merely to achieve a marginal lead in fitting a specific dataset distribution, but to enable the integration of heterogeneous data sources and the generation of scenes that exceed the complexity of the training distribution. While our FID/KID scores are comparable to global modeling methods like DiffuScene, **our method's true strength lies in its scalability and generalization**, as noted by reviewer GWbd. This capability is more clearly reflected in our qualitative results and user studies.
>
> ### **W4: Introduction writing and key challenges.**
>
> We appreciate the suggestion and will revise the introduction to explicitly highlight the three key challenges:
> 1.  **Data Scarcity:** Existing synthetic datasets (e.g., 3D-FRONT) are limited in scale and layout complexity, while real-world scans (e.g., ScanNet) contain noise that hinders direct conversion to high-quality layouts.
> 2.  **Global Modeling Constraints:** Current learning-based methods focus on global distributions, locking their capacity to the training set and preventing generalization to larger, more complex scenes.
> 3.  **LLM/VLM Spatial Deficits:** LLMs struggle with precise 3D coordinates and physical contact constraints, often leading to semantically inconsistent or physically floating layouts.
>
> ### **W5: Missing relevant references.**
>
> We will incorporate and discuss the suggested references in our revised version:
> While CasaGPT uses geometry-aware cuboid sequences to avoid collisions, it still focuses on fitting existing dataset distributions. Pair2Scene, conversely, models local interactions, allowing it to procedurally assemble scenes that go beyond the training data. Although global-local tree search uses regional generation, Pair2Scene focuses on fine-grained relations. By learning learned rules, our method generates more precise and diverse layouts compared to relying solely on VLM spatial priors.
>
> ### **Q1: Binary relations to multi-object configurations.**
>
> Complex configurations (e.g., four chairs around a table) are achieved through our local-to-global assembly process combined with collision-based rejection sampling. Through inference, we can obtain the spatial distribution of chairs (a multimodal distribution). When a sample is rejected due to a collision, we continue to sample its possible positions to simulate the global distribution. (more details in our response to Reviewer QpuK, Q3)
>
> ### **Q2: Determining the functional anchor.**
>
> For pairs with a clear primary-secondary hierarchy (e.g., a mouse and a computer), the primary object is designated as the anchor. For symmetric or equal relationships (e.g., two sofas), the LLM labels the relationship as equal, and during training, we randomly assign the anchor role to either object with equal probability to ensure balanced learning.
>
> ### **Q3: Internal support relationships.**
>
> Yes, our model can generate such configurations. Our dataset includes internal support surfaces. The geometry encoder captures these inner affordances, allowing the model to place objects inside the support anchor.
>
> We believe these clarify your concerns and will be included in the revision.

---

> > ### Author Rebuttal · Reviewer_QA5i · 2026-04-01
> >
> > Thanks for the authors for the detailed rebuttal.
> > The authors have fully resolved my concerns, I will change my rating to the **weak accept** score.
> >
> > This paper proposes a new insight in 3D scene generation: modeling the binary relationship.
> > In representation, the authors could carefully refine the writing, especially the introduction section.
> > The soundness concerns could be resolved by adding the rebuttal experiments in the final version.

---

### Official Review · Reviewer_GWbd · 2026-03-11

**Soundness:** 4
**Presentation:** 4
**Significance:** 4
**Originality:** 4
**Overall Recommendation:** 4
**Confidence:** 4

**Summary:**

This paper introduces Pair2Scene, a relational framework for 3D indoor scene generation that targets the challenges of scalability and limited data coverage. Instead of modeling an entire scene distribution directly, the method decomposes scene synthesis into localized procedural relations, enabling more effective modeling of complex spatial arrangements than prior global approaches. Built on the 3D-Pairs dataset and a Geometry-Aware Layout Predictor, Pair2Scene connects object geometry with spatial reasoning in a structured way. Experimental results show that the framework not only matches the training distribution well, but also produces dense scenes whose complexity goes beyond that seen in the training data. Overall, the work highlights relational modeling as a promising and scalable direction for learning-based scene generation.

**Compliance With Llm Reviewing Policy:**

Affirmed.

**Final Justification:**

The authors appropriately addressed most of my concerns, I would like to maintain my positive rating.

**Key Questions For Authors:**

1. The qualitative examples appear overly simplified, as the backgrounds are reduced to a flat floor plane. Real indoor environments typically involve more complex room structures and diverse layouts, making the generated scenes look toy-like and less realistic.

2. The paper does not clearly explain how the feasible set F is obtained in line291.

3. The definition of supporting relations appears limited to the case where one object is supported by another. It is unclear how the framework handles more complex cases involving mutual support between objects.

**Limitations:**

yes

**Strengths And Weaknesses:**

**Strengths**:
1. The motivation and problem formulation are good.
The paper proposes a clear and intuitive shift from global scene modeling to local relational modeling. By representing scenes through support and functional relations, the method introduces a more scalable inductive bias for dense indoor scene generation and directly addresses the difficulty of learning full-scene joint distributions.

2. Substantial technical and dataset contribution.
Beyond the model itself, the paper contributes a structured relational representation and a data curation pipeline that builds the 3D-Pairs dataset, containing about 140K relation tuples collected from multiple sources. This makes the work more complete and increases its potential value.

3. Strong empirical support for generalization beyond training complexity.
The experiments suggest that Pair2Scene can not only fit the target data distribution, but also generate denser and more complex scenes, which is important.

**Weakness**:
1. Physical affordance modeling. In line199-200, the use of a geometry encoder alone to capture physical affordance appears insufficient. Physical affordance is not determined solely by 3D geometry, but also depends on other properties such as material characteristics and surface attributes.

2. Equation 2 models the probability of the 12 bounding-box parameters independently, which is not reasonable. Since object spatial configuration is jointly defined by the full bounding box, these parameters should be modeled in a coupled rather than factorized manner.

3. Global scene consistency. Although local generation can be iteratively composed into a large scene, this procedure does not explicitly account for relations across different local regions, which may compromise the semantic coherence of the overall scene.

---

> ### Author Rebuttal · Authors · 2026-03-31
>
> Thank you sincerely for your valuable time and insightful comments.
>
> ### **W1: Physical affordance modeling is insufficient.**
>
> We agree that material properties play a role in physical interactions, yet in the majority of indoor scenes, the geometric structure can determine the semantic relationships and spatial placement. Our geometry encoder is specifically designed to effectively capture these dominant spatial constraints, and since the framework is inherently modality-agnostic, incorporating surface attributes or material embeddings into the encoder represents a straightforward extension that can be explored in future work to further refine physical realism.
>
> ### **W2: Bounding box parameters are modeled independently.**
>
> The independent factorization in Eq. 2 is a design choice to simplify the pipepine and ensure computational efficiency. This approach is consistent with existing methods, such as ATISS, which also independently models the parameters of logistic distribution across different dimensions. Additionally, all parameters are predicted from a shared joint feature representation, allowing the model to implicitly capture correlations. Our qualitative results demonstrate that this does not limit the model’s ability to generate coherent and realistic bounding box configurations.
>
> ### **W3: Lack of explicit global scene consistency.**
>
> Our core insight is that global layouts are composed of independent local relational compositions, a local-to-global approach that is precisely what allows Pair2Scene to scale up and generate scenes exceeding the complexity of the training distribution. In some corner cases, our method is hard to handle the complex dependencies between multiple objects. However, to ensure global coherence, we employ collision-based rejection sampling to simulate global spatial constraints, and we have added new ablation studies to our **[rebuttal material](https://anonymous-for-content-submission.github.io/rebuttal-material/)** showing the effect of rejection sampling. Further, our qualitative results demonstrates that our method can maintain high semantic consistency across complex layouts in most cases, such as four chairs around a dining table.
>
> ### **Q1: Generated scenes look toy-like due to simplified results.**
>
> The simplified results are primarily a reflection of the limitations of existing 3D scene datasets. Most furniture layouts in our results are learned from 3D-FRONT, which inherently features simplified indoor scene. Our primary objective is to demonstrate the capability to generate dense, complex object distributions that go beyond the training data and integrate data from various sources. We believe that as more diverse and structurally complex datasets become available, our pipeline will naturally produce more realistic environments without requiring architectural changes.
>
> ### **Q2: Explanation of the feasible set $F$ in line 291.**
>
> We clarify that the feasible set $F$ is determined via collision detection. Specifically, a generated object configuration is considered to be within $F$ if it does not result in physical intersections with existing objects or room boundaries. We will update the manuscript to make this definition explicit.
>
> ### **Q3: Handling of complex/mutual support relations.**
>
> Based on our analysis of indoor datasets, the vast majority of objects are supported by a single primary surface, leading us to focus our current framework on this dominant support mapping. In rare cases involving mutual support, such as two books leaning against each other, these can be treated as a single semantic group where the shelf remains the primary support for this group.
>
> We believe these clarify your concerns and will be included in the revision.

---

> > ### Author Rebuttal · Reviewer_GWbd · 2026-04-03
> >
> > Thanks for the authors for the detailed rebuttal. The authors have fully resolved my concerns, and I will maintain my positive rating

---

### Decision · Program_Chairs · 2026-04-30

**Decision:**

Accept (regular)

**Comment:**

This paper received positive reviews with scores of 4, 4, 4, and 4. Two reviewers acknowledged that the rebuttal fully addressed their concerns, while the other two felt their concerns were partially addressed. Nevertheless, all reviewers recommended a "weak accept."

Reviewer QA5i acknowledged the novelty of the proposed approach for generating large 3D scenes and suggested that the final version be carefully refined in its writing, and it should also include the experimental evidence provided in the rebuttal.

Reviewer QpuK requested that the abstract and claims should be revised for clarity, and an ablation study on rejection sampling and gravity should be included. Furthermore, LLM and tree generation processes need to be more clearly explained.

Reviewer 1csW acknowledged the value of the system-level contribution and appreciated the effort made by the authors in providing additional rejection target analysis and paired t-tests, which help clarify the model's design intent. However, the reviewer also requested improvements to the paper's framing and raised concerns about the limited methodological novelty of some individual components.

The area chair concurs with the reviewers' positive assessments and recommends the acceptance of this paper.